# ThinKV: Thought-Adaptive KV Cache Compression for Efficient Reasoning Models

**Akshat Ramachandran**[1†]**, Marina Neseem**[2]**, Charbel Sakr**[2]**, Rangharajan Venkatesan**[2]**,
Brucek Khailany**[2]**, and Tushar Krishna**[1]

[1]Georgia Institute of Technology, Atlanta, USA
[2]NVIDIA Research, Santa Clara, USA
[1]akshat.r@gatech.edu, tushar@ece.gatech.edu
[2]{mneseem, csakr, rangharajanv, bkhailany}@nvidia.com
†Work done during an internship at NVIDIA

## ABSTRACT

The long-output context generation of large reasoning models enables extended chain of thought (CoT) but also drives rapid growth of the key–value (KV) cache, quickly overwhelming GPU memory. To address this challenge, we propose ThinKV [1], a thought-adaptive KV cache compression framework. ThinKV is based on the observation that attention sparsity reveals distinct thought types with varying importance within the CoT. It applies a hybrid quantization–eviction strategy, assigning token precision by thought importance and progressively evicting tokens from less critical thoughts as reasoning trajectories evolve. Furthermore, to implement ThinKV, we design a kernel that extends PagedAttention to enable efficient reuse of evicted tokens' memory slots, eliminating compaction overheads. Extensive experiments on DeepSeek-R1-Distill, GPT-OSS, and NVIDIA AceReason across mathematics and coding benchmarks show that ThinKV achieves near-lossless accuracy with less than 5% of the original KV cache, while improving performance with up to $5.8\times$ higher inference throughput over SoTA baselines.

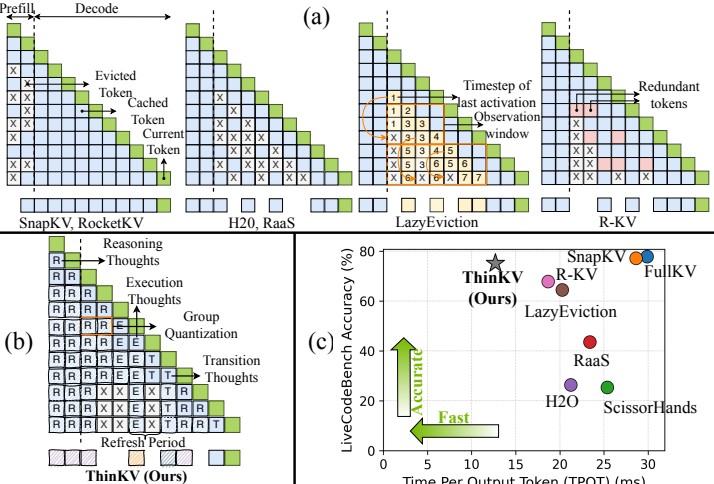

Figure 1: Illustrative comparison of KV cache compression methods as tokens are generated: (a) Existing techniques: SnapKV, RocketKV, H2O, RaaS, LazyEviction and R-KV, and (b) **ThinKV (Ours)**. (c) Accuracy vs. TPOT comparison for GPT-OSS-20B evaluated on LiveCodeBench.

## 1 INTRODUCTION

Long-context modeling (Yuan et al., 2025) is a core capability for next-generation LLMs. While early work focused on long-input contexts, the advent of **Large Reasoning Models (LRMs)**—e.g., OpenAI's O-series (OpenAI, 2024) and DeepSeek-R1 (Guo et al., 2025)—has shifted 'attention' to

---

[1]The term may be interpreted either as *Thin* KV or as *Think* KV

**long-output contexts**, involving generation of thousands of tokens (Liu et al., 2025). This capability facilitates extended reasoning (Zhu et al., 2025) and long-horizon code generation (Seo et al., 2025).

LRMs attain state-of-the-art reasoning accuracy by generating long chains of thought (CoT) (Wei et al., 2022), producing extended rationales to explore and verify solutions. However, long CoT generation incurs substantial memory overheads due to rapid growth of the key–value (KV) cache during decoding (Li et al., 2025). In code generation (Jain et al., 2024), for instance, a GPT-OSS-20B (Agarwal et al., 2025) producing ∼**32K tokens** with batch size 32 requires 50 GB for the KV cache and 40 GB for weights—exceeding the 80 GB of an NVIDIA A100. Since the decode stage is memory-bound (Recasens et al., 2025), the KV cache becomes the central bottleneck for long-output context generation. KV cache compression thus offers a promising solution.

## 1.1 RELATED WORK AND LIMITATIONS OF EXISTING COMPRESSION TECHNIQUES

Existing compression approaches span quantization, eviction, low-rank approximation, and hybrids thereof. Most, however, focus on the prefill phase of long-input tasks (Li et al., 2024) (Figure 1(a)) and are ill-suited for LRMs and long-output generation. A few works study decode-stage compression (Zhang et al., 2023; Shi et al., 2025; Liu et al., 2024b) for LLMs, but typically use greedy recency-based eviction (Figure 1(a)) or uniform quantization, both of which overlook reasoning dynamics, leading to degraded LRM accuracy (Figure 1(c)). For additional details refer §B.

**LRM KV Cache Compression.** Recent work has moved beyond simple recency-based eviction towards methods that partially capture reasoning dynamics (Figure 1(a)). RaaS (Hu et al., 2025) preserves tokens with re-emergent importance to avoid premature eviction; LazyEviction (Zhang et al., 2025a) delays eviction to retain tokens likely to recur by tracking attention activity; R-KV (Cai et al., 2025) combines attention-based importance with redundancy; and PM-KVQ (Liu et al., 2025) progressively reduces token precision during decoding. However, they operate at the token level, making compression decisions that overlook the broader semantic structure of reasoning. This can cause removal of reasoning-critical tokens or limit compression by overweighting less important ones, yielding suboptimal accuracy–efficiency trade-offs (Figure 1(c)), under high compression.

**System.** Dynamic token eviction creates memory holes, causing internal fragmentation (Kwon et al., 2023). H2O (Zhang et al., 2023) mitigates this with circular buffers, but these support only contiguous eviction, whereas LRM policies conduct non-contiguous token removal. While other methods (Cai et al., 2025) explore gather-based compaction; it requires irregular, index-based memory accesses that contend heavily for HBM bandwidth. Our analysis (§5.1) reveals that gather, sharply increases time per output token (TPOT) (Figure 1(c)), consistent with Kwon et al. (2023).

## 1.2 CONTRIBUTIONS

Motivated by these limitations, we ask: *Can a KV cache compression framework go beyond token-level heuristics to preserve reasoning-critical information under high compression while maximizing efficiency?* We present **ThinKV** (Figure 1(b)), a **thought-adaptive** hybrid quantization–eviction framework (§2) with four key components:

- **Thought Decomposition (§3.1, §4.1):** We show the CoT in LRMs can be decomposed into distinct thought types, with their differentiation enabled by degree of sparsity in attention weights.
- **Think Before you Quantize (TBQ) (§3.2, §4.2):** We propose a KV cache quantization scheme that assigns precision to tokens based on the importance of their associated thought type.
- **Think Before You Evict (TBE) (§3.3, §4.3):** We introduce TBE, a thought-adaptive eviction scheme that leverages inter-thought dynamics to progressively evict tokens.
- **Continuous Thinking (§5):** We design a kernel extending PagedAttention that efficiently reuses evicted memory slots for subsequent tokens without relying on expensive compactions.

Through algorithm–system co-design, ThinKV delivers aggressive KV cache compression while preserving accuracy and improving inference efficiency (§6). On mathematics and coding benchmarks with DeepSeek-R1-Distill-Llama, GPT-OSS, and several other LRMs, ThinKV achieves **near-lossless accuracy with under 5% of the original KV cache**, outperforming state-of-the-art baselines with up to **1.68× lower TPOT** (Figure 1(c)) and up to **5.80× higher throughput**.

## 2 WHY QUANTIZATION+EVICTION ?

The memory footprint of the KV cache can be expressed as $\mathrm{Mem}(KV) \propto (I + bL_{\mathrm{gen}}) \times a\beta$, where $I$ is the prompt length, $L_{\mathrm{gen}}$ the total number of generated tokens, $\beta$ denotes the bytes per parameter. The factors $a, b \in [0, 1]$ capture memory reductions from quantization and eviction, respectively. Uncompressed KV cache corresponds to $a = 1$ (full precision) and $b = 1$ (no eviction).

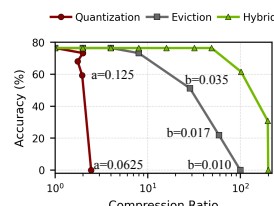

For quantization (Ramachandran et al., 2024b), we adopt KIVI (Liu et al., 2024b) as the representative. As shown in Figure 2 (GPT-OSS 20B on LiveCodeBench), reducing $a$ fails to proportionally increase compression ratio, since in LRMs we find that aggressive quantization inflates $L_{\mathrm{gen}}$, eroding memory savings and simultaneously degrading accuracy. Under eviction–using R-KV (Cai et al., 2025)–reducing $b$ initially increases compression ratio while preserving accuracy. Unlike quantization, eviction does not cause an increase in generation length; however, as $b \to 0$, accuracy degrades with higher compression.

Figure 2: Accuracy-compression tradeoff of quantization, eviction and hybrid approaches.

Hybrid compression (ThinKV,§4) traces a Pareto-optimal frontier, sustaining high accuracy at much higher compression ratios. We believe that by combining quantization and eviction, it partially regularizes quantization-induced length inflation and maintains accuracy at extreme compression by flexibly trading off token count and precision.

## 3 MOTIVATING ANALYSES

In this section, we present three key observations that motivate the design of ThinKV's algorithm.

### 3.1 ATTENTION SPARSITY FOR DYNAMIC THOUGHT DECOMPOSITION

**Definition 1** (LRM Thought Decomposition). *Let $\mathcal{T} = \{c_0, c_1, \ldots, c_{|\mathcal{T}|-1}\}$ denote the set of thought categories. During generation, an $L$ layer LRM produces a sequence $(y_0, \ldots, y_{n-1})$, where each $y_i$ is a discrete token. At decoding step $i$, the cache of layer $\ell$ is denoted by $S_i^\ell$, representing the set of stored KV pairs up to that point. Thought decomposition is defined as,*

- *For each step $i \in [n]$ and generated token $y_i$, a categorization function associates a category label $c_j$ for $j \in [|\mathcal{T}|]$ as, $\phi : \{y_0, \ldots, y_{n-1}\} \to \mathcal{T}, \ \phi(y_i) = c_j$.*
- *Each token generates one KV entry per layer, which is assigned a category as identified above. Formally, $S_i^\ell \setminus S_{i-1}^\ell = \{(K_i^\ell, V_i^\ell, c_j)\}$, where the KV entry is associated with its thought type $c_j$.*

An exact realization of $\phi$ is nontrivial. Prior works approximate $\phi$ by maintaining a keyword list for each category; Venhoff et al. (2025) found $|\mathcal{T}|$=8 categories, while Chen et al. (2025b) identified $|\mathcal{T}|$=3. However, keyword-based methods fail when models generate lexical variations and tokens outside keyword lists (Agarwal et al., 2025).

We present an empirical observation that enables a generalizable approximation of $\phi$, based on the sparsity pattern Ramachandran et al. (2025b) of the normalized attention scores[2]. Figure 3 reports layer-wise sparsity ratios for two LRMs (R1-Llama-8B and -70B) on AIME and LiveCodeBench prompts respectively. We draw the following key observations:

**Observation 1a:** The attention sparsity pattern across decode steps exhibits a tri-modal distribution.

To *only* interpret the sparsity regions, we follow Chen et al. (2025b) and assign representative keywords (§D.2) as illustrative labels. This categorization yields three thought types

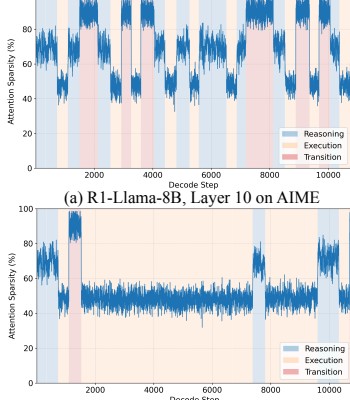

(a) R1-Llama-8B, Layer 10 on AIME

(b) R1-Llama-70B, Layer 4 on LiveCodeBench

Figure 3: Layer-wise attention sparsity across decode steps for R1-Llama-8B on AIME and R1-Llama-70B on LiveCodeBench.

---

[2]The normalized attention scores are defined as $\mathrm{softmax}(qK^\top)$ and sparsity is measured by setting a threshold at $1\%$ of the row-wise maximum, following Zhang et al. (2023).

($|\mathcal{T}| = 3$): **reasoning (R)**, involving systematic thinking; **execution (E)**, encompassing calculations, or code generation; and **transition (T)**, capturing uncertainty and backtracking behavior.

**Observation 1b:** $\mathcal{T} = \{R, T, E\}$, with **T** thoughts exhibiting the highest sparsity, followed by **R** thoughts, while **E** thoughts have the lowest sparsity.

Some layers exhibit more than three sparsity regimes or ambiguous boundaries (§E.4). As shown in §6, fixing $|\mathcal{T}| = 3$ and choosing the optimal layer subset $\mathcal{L}^*$ achieves the best accuracy.

## 3.2 LRM THOUGHT IMPORTANCE

We examine relative thought importance as the basis for our thought-adaptive quantization scheme. Consider an LRM CoT output consisting of $N$ thought segments ($Y_i$), followed by a final answer $A$. Inspired by Bogdan et al. (2025), we measure the counterfactual importance of each segment $Y_i$ by computing the KL divergence between $A$'s distributions obtained with and without $Y_i$, averaged over 50 rollouts. Figure 4 presents thought importance for GPT-OSS-20B on AIME and Live-CodeBench.

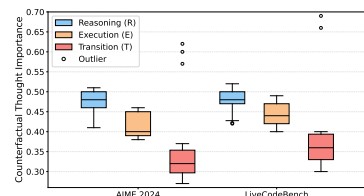

Figure 4: Counterfactual importance of thought categories for GPT-OSS-20B on AIME and LiveCodeBench.

**Observation 2.** We observe a clear hierarchy of thought importance: $\mathbf{R} > \mathbf{E} > \mathbf{T}$. Interestingly, we find outlier T thoughts with unusually high importance which correspond to backtracking behavior and removing them causes the model to loop endlessly (see example in §E.17).

## 3.3 LRM THOUGHT ASSOCIATION

We analyze inter-thought dynamics by measuring pairwise associations (Bogdan et al., 2025). For $(Y_i, Y_j), j > i$, we suppress attention to $Y_i$ (all layers and heads) and compute the KL divergence of $Y_j$'s logits, averaging over its tokens to obtain a directed association score, indicating the extent $Y_j$ depends on $Y_i$. Figure 5 illustrates the influence of thought $Y_i$ (X-axis) on subsequent thoughts $Y_j$ (Y-axis) during generation for an AIME prompt (additional visualizations in §E.5).

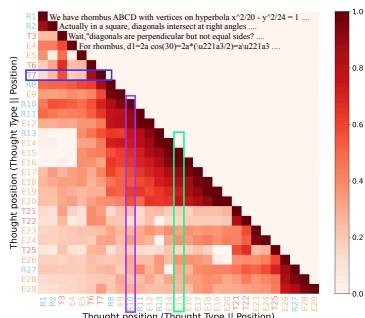

Figure 5: Pairwise thought associations for GPT-OSS-20B on AIME. $c_j\alpha$ denotes thought segment type and its position in CoT.

**Observation 3.** With every **T** thought, all prior thought segments become progressively less influential (fewer tokens need to be retained), underscoring its role in altering the reasoning trajectory. Note **R** and **E** segments highlighted with ☐ and ☐, respectively. Additionally, T thoughts are weakly influenced by prior context (high sparsity) (☐), while E thoughts depend heavily on context bounded between consecutive transitions (low sparsity), bolstering *Observation 1b*.

## 4 THINKV METHODOLOGY

In this section, we present ThinKV's hybrid scheme, which first decomposes tokens into distinct thought types (§4.1) and then applies thought-adaptive quantization (§4.2) and eviction (§4.3).

## 4.1 ATTENTION SPARSITY GUIDED CONSTRUCTION OF $\phi$

Building on the observations in §3.1, we now detail how ThinKV leverages attention sparsity to dynamically identify thought types, forming the basis of it's adaptive compression strategy.

**Offline Calibration.** We use kernel density estimation (KDE) (Parzen, 1962) to derive the $|\mathcal{T}| - 1$ sparsity thresholds $\Theta = \{\theta_1, \ldots, \theta_{|\mathcal{T}|-1}\}$ that separate thoughts. From a calibration set of $P$ prompts, we estimate KDE per prompt and select the layer subset $\mathcal{L}^*$ that exhibits $|\mathcal{T}|$ modes. We extract $|\mathcal{T}|-1$ thresholds by identifying local minima between modes (statistical term), and compute final thresholds by averaging across all prompts and layers in $\mathcal{L}^*$. Refer §D.1 for algorithm.

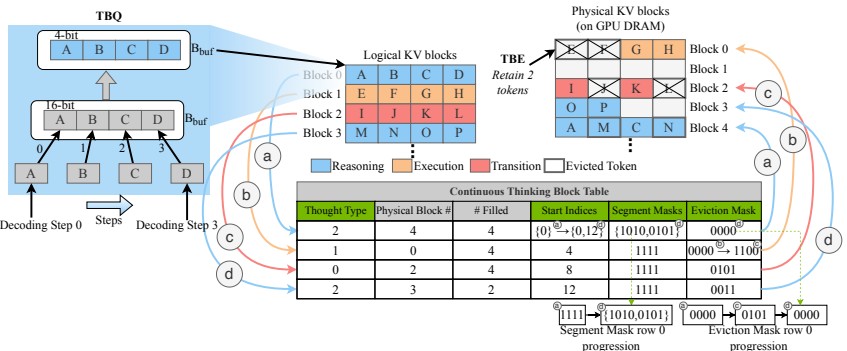

Figure 6: Walkthrough of ThinKV with $\tau = g =$ block size $= 4$, $R = \{2\}$.

**Decode-Time Behavior.** During generation, $\phi$ is approximated by averaging sparsity over $\mathcal{L}^*$ and comparing with thresholds $\Theta$ to determine the thought type. From Figure 3 and consistent with Chen et al. (2025b), thought segments[3] in the CoT typically span 100–300 tokens. We therefore set a refresh interval of $\tau = 128$ steps, updating categories only at these intervals to minimize overhead.

## 4.2 THINK BEFORE YOU QUANTIZE (TBQ)

**Problem Formulation 1** (Thought-Adaptive Quantization). *Let $\mathcal{B} = \{b_0, b_1, \ldots, b_{|\mathcal{T}|-1}\}$ denote the set of available quantization bit-precisions, ordered such that $b_0 < b_1 < \cdots < b_{|\mathcal{T}|-1}$. We define a KV cache quantization policy that allocates precision to tokens according to thought importance:*

- *Define an importance function $\rho : \mathcal{T} \rightarrow \mathbb{N}$ that assigns each thought type $c_j \in \mathcal{T}$ a score $\rho(c_j)$. We then construct a mapping $\psi : \mathcal{T} \rightarrow \mathcal{B}$ such that higher importance implies higher precision, i.e., $\rho(c_{j_1}) > \rho(c_{j_2}) \Rightarrow \psi(c_{j_1}) \geq \psi(c_{j_2})$.*
- *Each new KV entry $(K_i^\ell, V_i^\ell, c_j) \in S_i^\ell \setminus S_{i-1}^\ell$ is quantized with bit-precision $\psi(c_j)$, yielding $(\tilde{K}_i^\ell, \tilde{V}_i^\ell, c_j)$, where $\tilde{K}_i^\ell, \tilde{V}_i^\ell$ denote the quantized KV representations.*

Building on the observed thought importance in §3.2, $\rho(R) = 2, \rho(E) = 1, \rho(T) = 0$. We construct $\mathcal{B} = \{2, 4, 8\}$ with ternary for 2-bit, NVFP4 (Alvarez et al., 2025) for 4-bit, and FP8 for 8-bit. Ternary and NVFP use group quantization with $g = 16$ and a shared FP8 (E4M3) scale factor, whereas FP8 employs a per-tensor FP32 scale factor (see §D.3). We assign **R**, **E**, and **T** thought tokens to 8-, 4-, and 2-bit precision, respectively. Notably, as shown in §6, **R** tokens maintain comparable accuracy even at 4-bit, allowing adoption of 4-bit for **R** in practice without loss of performance. Following Liu et al. (2024b), keys are quantized per-channel while values are quantized per-token. A buffer $B_{\text{buf}}$ of size $g$ stores tokens in full precision until the group size is reached, after which they are group quantized. Figure 6 (TBQ) presents an example with $g = 4$.

## 4.3 THINK BEFORE YOU EVICT (TBE)

**Problem Formulation 2** (Thought-Adaptive Eviction). *Let $k$ be the cache budget, $S_i^\ell(c_j) \subseteq S_i^\ell$ be the KV entries of a thought segment of type $c_j$ and $\mathcal{R} = \{R_0, R_1, \ldots, R_{m-1}\}$ denote the set of $m$ retention rates, in descending order, where $R_n$ specifies the number of tokens to be preserved when a segment is selected for eviction the $n$-th time. Eviction policy $\pi : S_i^\ell(c_j) \mapsto S_i^{\ell*}(c_j)$ is defined as,*

- ***Case 1:*** *If a reasoning trajectory-changing thought $c_t$ is generated, $\pi$ progressively evicts preceding thoughts such that $|S_i^{\ell*}(c_j)| = \min(|S_i^\ell(c_j)|, R_n)$, where $n$ identifies number of times preceding thought $c_j$ has been selected for eviction (i.e., the number of trajectory changes in reasoning).*
- ***Case 2:*** *If no $c_t$ thoughts are generated, but $|S_i^\ell| > k$, we find the oldest and least important thought segment to apply $\pi$ until $|S_i^{\ell*}| \leq k$.*

Following from the observation in §3.3, transition thoughts are the reasoning trajectory-changing thoughts $c_t$. Since we employ a refresh period of $\tau = 128$, every thought segment contains 128 tokens. Therefore, following Problem Formulation 2, we define the retention schedule as $\mathcal{R} = \{64, 32, 16, 8, 4\}$ for all thought types, with a minimum retention of 4 tokens per segment

---

[3]A contiguous span of tokens assigned to the same thought type.

(see Figure 11(a)). At each transition thought $c_t$, the eviction policy $\pi$ anneals preceding segments (including previous transitions) by reducing them to the next lowest retention level in $\mathcal{R}$ (see Figure 6). With successive transitions, all previous thought segments are progressively shrunk until the minimum retention value is reached. If no $c_t$ occurs or all segments before the current $c_t$ are already at their minimum, $\pi$ evicts from the oldest and least important segment to its next lowest retention level in $\mathcal{R}$. TBE is a proactive eviction scheme that operates at the granularity of thought segments, evicting large sets of low-importance tokens as opportunities arise rather than waiting for cache saturation and stepwise per-token removal. This strategy reduces eviction frequency and, as shown in §6, minimizes overhead.

**Eviction Policy ($\pi$).** We apply K-means clustering to post-RoPE key embeddings (He et al., 2025), with K= $\min(|S_i^\ell(c_j)|, R(m, c_j))$. The cluster centroids correspond to keys that are retained, and the corresponding value tokens are preserved. An illustration is provided in §D.4.

## 5 THINKV SYSTEM IMPLEMENTATION

We introduce *Continuous Thinking* (CT), an extension of PagedAttention (Kwon et al., 2023) to enable in-place memory reuse of evicted KV tokens, without expensive gather-based compactions.

### 5.1 THE COST OF GATHER-BASED COMPACTION

Existing LRM eviction methods drop non-contiguous tokens from arbitrary positions within the CoT, causing internal fragmentation that requires gather-based compaction. To quantify its overhead, we study R-KV Cai et al. (2025) with a 1024-token budget. We implement two Triton gather kernels: (a) a sequential variant and (b) an overlapped variant employing separate CUDA streams to run concurrently. Figure 7 reports kernel performance on DeepSeek-R1-Distill-Llama-8B.

**Observation 4a (Sequential).** Per-layer gather overhead grows sharply with batch size (Figure 7(a)), causing up to $37\times$ TPOT slowdown.

**Observation 4b (Overlapped).** At small batch sizes, the gather cost is effectively hidden, yielding lower TPOT relative to the sequential case. As batch size grows, however, overlapped gather begins to interfere with subsequent-layer's attention, as shown in Figure 7(b). Specifically, contention arises on HBM bandwidth, where the gather kernel's KV writes conflict with the attention kernel's KV reads. This contention inflates attention time (up to ~35% slowdown), and thus causes higher TPOT.

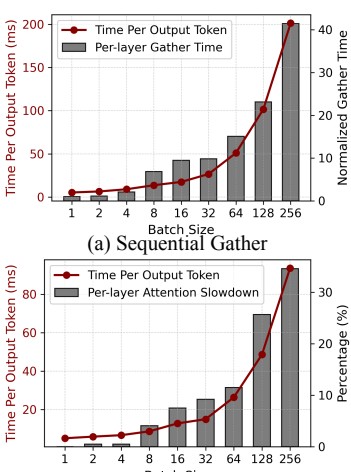

(a) Sequential Gather

(b) Overlapped (Parallel) Gather

Figure 7: Performance of sequential and overlapped gather kernel on R1-Llama-8B.

### 5.2 CONTINUOUS THINKING (CT)

**Block Table.** PagedAttention maintains a block table for each request and each layer. Figure 6 (see §D.6 for detailed walkthrough) shows the modified block table, recording the following information (new fields in green),

• *Physical block #* and *# Filled*: KV block index in GPU memory and its token count.
• *Thought type*: Thought type of tokens in a block; CT implements thought-aware paging.
• *Start indices*: Records the start position of the thought segment of tokens in the physical block.
• *Segment masks:* If there are multiple start indices, the segment mask is a bit vector (length=block size) that marks the locations corresponding to each start index with a 1.
• *Eviction mask*: A bit vector marking positions of tokens evicted by TBE with 1s.

**TBE with CT.** The CT kernel collaborates with TBE to perform eviction. As shown in Figure 6, TBE selects segments for progressive eviction using the *thought type* and *start index* fields. Tokens marked for eviction are not immediately removed; instead, they are soft-marked in the *eviction mask*, with actual removal deferred until new tokens arrive to overwrite into the evicted slots.

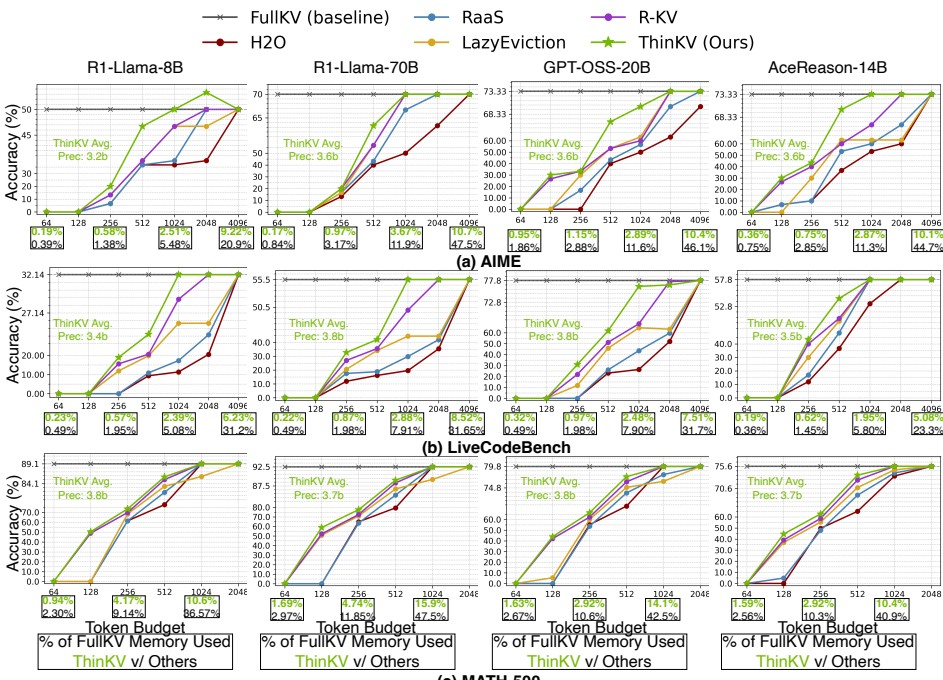

Figure 8: ThinKV compared with SoTA eviction baselines, reported as pass@1 accuracy.

**Efficient Memory Management.** When new tokens of a thought type are generated, the CT kernel uses the *eviction mask* to identify reclaimable slots in existing blocks of the same type. The *start index* of the new thought segment is appended to the existing block table entry, and the *segment mask* updated to mark its token positions. By reusing slots in place, CT avoids compaction and eliminates fragmentation. Moreover, tokens need not be reordered during attention computation, since attention is permutation-invariant (§C.3). Therefore, our modifications leave the PagedAttention kernel for attention computation unchanged enabling seamless integration with serving frameworks.

## 6 EVALUATION

### 6.1 EXPERIMENTAL SETUP

**Models and Datasets.** We evaluate on DeepSeek-R1-Distill-Llama (8B and 70B), DeepSeek-R1-Distill-Qwen-14B, GPT-OSS (20B and 120B), QwQ-32B, AceReason-Nemotron-14B, and MobileLLM-R1-950M. Evaluations span mathematics (MATH-500 (Lightman et al., 2023), AIME (MAA, 2024), GSM8K (Cobbe et al., 2021)) and code generation (LiveCodeBench (Jain et al., 2024)). For calibration, we randomly sample 100 prompts from s1K (Muennighoff et al., 2025).

**Hyperparameters.** We set number of thoughts $|\mathcal{T}| = 3$, optimal calibration layers $|\mathcal{L}^*| = 4$, refresh rate $\tau = 128$, group size $g = 16$, retention rates $\mathcal{R} = \{64, 32, 16, 8, 4\}$ and CT block size = 8. **R** and **E** thoughts are quantized to 4-bits and **T** thoughts to 2-bits.

**Baselines.** We compare accuracy against eviction baselines, H2O (Zhang et al., 2023) (LLMs) and RaaS (Hu et al., 2025), R-KV (Cai et al., 2025), LazyEviction (Zhang et al., 2025a) (LRMs), as well as quantization baselines, KIVI (Liu et al., 2024b) (LLMs) and PM-KVQ (Liu et al., 2025) (LRMs).

**System Optimizations.** We implement ThinKV in a hardware-friendly manner for GPUs. We design optimized CUDA kernels for group quantization and following Liu et al. (2024b), we fuse dequantization with matrix multiplication to reduce overhead. Two T tokens at 2-bits are packed into a 4-bit format, consistent with R/E tokens, ensuring aligned memory. TBE's K-means–based eviction is accelerated on GPUs with CUDA, following Kruliš & Kratochvíl (2020). CT is fully implemented in Triton, extending the PagedAttention kernel of OpenAI (2025).

**Evaluation Setup.** All experiments are conducted on $1\times$NVIDIA A100 80GB GPU and $1\times$ NVIDIA GH200 Superchip. Following Cai et al. (2025), we constrain maximum generation length to $32K$ tokens. For accuracy evaluation, for each question, we generate 8 independent responses

and compute pass@1 accuracy as pass@1 $= \frac{1}{k}\sum_{i=1}^{k} p_i$, where $p_i$ denotes whether the $i$-th sampled response is correct Ramachandran et al. (2024a). All throughput and latency numbers are obtained by averaging across 3 independent runs. Importantly, in our experiments we treat prefill-tokens as **R** type (see Figure 1(b)). Refer Appendix E for additional details.

## 6.2 MAIN RESULTS

**Accuracy Comparison with Eviction Baselines.** In Figure 8, we evaluate diverse LRM families on reasoning datasets with KV cache budgets ranging from 64 to 4096 tokens. The average generation lengths are 9,020 tokens on AIME, 14,166 on LiveCodeBench, and 2,468 on MATH-500. On challenging reasoning benchmarks such as AIME and LiveCodeBench, **ThinKV achieves competitive accuracy with a cache budget of 1024 tokens, accounting for $< 3.67\%$ of FullKV memory**, whereas other methods require $> 12\%$ to reach similar accuracy. For R1-Llama-8B and AceReason-14B on AIME, ThinKV sustains $< 4\%$ **drop using only $\sim 1.3\%$ of the KV cache**. ThinKV's hybrid quantization–eviction and thought-adaptive scheme, enables superior accuracy while sustaining higher compression. ThinKV operates at an average precision of 3.4 bits, with harder problems achieving lower precision due to more frequent transition thoughts.

Table 1: Comparison of ThinKV with KV quantization baselines.

| Model | Method | Bit-Width | AIME | LiveCodeBench |
|---|---|---|---|---|
| R1-Qwen-14B | Baseline | 16-16 | 53.33 | 47.90 |
| | KIVI | 2-2 | 40.00 | 34.56 |
| | PM-KVQ | 3.2-3.2 | 43.33 | 41.97 |
| | **ThinKV (k=1024)** | 3.5-3.5 | **50.00** | **45.84** |
| QwQ-32B | Baseline | 16-16 | 73.33 | 55.45 |
| | KIVI | 2-2 | 60.56 | 40.75 |
| | PM-KVQ | 3.5-3.5 | 67.86 | 46.68 |
| | **ThinKV (k=1024)** | 3.4-3.4 | **70.28** | **50.47** |

Table 2: Throughput (tokens/s) comparison on GPUs. *Mem. ftprnt: Memory footprint (%) normalized to FullKV.

| Method | Tok. Budget | Mem. ftprnt (%)* | A100 Batch | A100 Tok/s | GH200 Batch | GH200 Tok/s |
|---|---|---|---|---|---|---|
| FullKV | – | 100 | 13 | 297.5 | 19 | 453.9 |
| R-KV (seq) | 1024 | 5.48 | 268 | 1450.5 | 350 | 2425.8 |
| R-KV (ovl) | 1024 | 5.48 | 268 | 2320.9 | 350 | 4311.3 |
| **ThinKV** | 1024 | 2.51 | **711** | **8412.2** | **938** | **10578.5** |
| *Iso-batch, Iso-compression comparison* | | | | | | |
| R-KV (seq) | 1024 | 5.48 | 256 | 1769.3 | 256 | 2489.8 |
| R-KV (ovl) | 1024 | 5.48 | 256 | 3539.3 | 256 | 5318.7 |
| **ThinKV w/o TBQ** | 1024 | 5.78 | 256 | **5298.4** | 256 | **8079.9** |

**Accuracy Comparison with Quantization Baselines.** We summarize our findings in Table 1, using $k = 1024$ for ThinKV. KIVI applies uniform INT quantization across all tokens, while PM-KVQ progressively reduces precision to a final 2-bit representation. Both approaches treat all tokens as equally important, leading to substantial accuracy degradation on LRMs. In contrast, ThinKV's thought-adaptive quantization (TBQ) assigns precision based on thought-type importance, achieving **minimal accuracy loss with an average precision of 3.4 bits**.

Table 3: ThinKV throughput on R1-Llama-8B (A100-80GB, 32K generation) with 2048 token budget.

| Method | Acc. | Batch Size (max) | Token Budget | Throughput |
|---|---|---|---|---|
| FullKV | 50 | 13 | – | 297.5 |
| **ThinKV** | **50** | **290** | **2048** | **4688.4** |

**Throughput Analysis.** Table 2 reports end-to-end throughput on two GPUs for a R1-Llama-8B performing continuous generation of $32K$ tokens. As baselines, we include two R-KV variants: one performing sequential gather (seq) and the other overlapped gather (ovl). FullKV and R-KV use FlashAttention (Dao, 2023), while ThinKV employs the CT kernel. For each method, we report the maximum batch size achievable on different GPUs. At batch size 1, all techniques achieve comparable performance with only marginal improvements over FullKV (Cai et al., 2025). The main throughput gains come from **ThinKV's ability to sustain larger and more efficient batch inference**. Specifically, ThinKV's hybrid scheme attains a higher compression ratio, supporting up to $3\times$ **larger batch sizes than R-KV and yielding throughput gains of up to** $5.8\times$ **over R-KV (seq) and** $3.6\times$ **over R-KV (ovl)**. To isolate CT kernel's impact on ThinKV throughput at larger batch sizes, we conduct an iso-batch, iso-compression (ThinKV w/o TBQ) comparison with a batch size=256. **ThinKV achieves up to** $3.2\times$ **and** $1.6\times$ **higher throughput** than R-KV (seq) and R-KV (ovl), respectively, due to the elimination of gather-based compaction.

In Table 2, we report results using a 1024-token budget with R4E4T2 precision, which limits accuracy loss to $\leq 1\%$ for most LRMs and datasets. For more sensitive models, we also evaluate a conservative 2048-token budget that preserves accuracy across all settings. As shown in Table 3, ThinKV at 2048 tokens increases the maximum batch size from **13 to 290** and achieves a $15.8\times$ throughput gain over FullKV, demonstrating substantial acceleration under accuracy-preserving constraints.

**E2E System Throughput versus User Latency Analysis.** Motivated by the dynamic-serving analyses in Kwon et al. (2023); Yu et al. (2022), we evaluate ThinKV under multi-user concurrency.

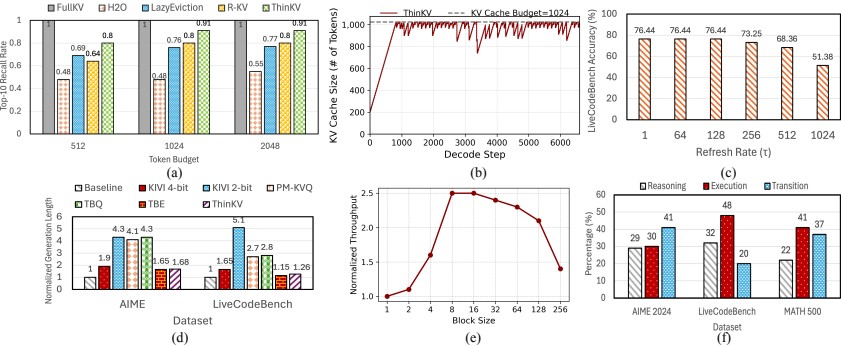

Figure 10: ThinKV ablation experiments: (a) recall rate of tokens with Top-10 attention scores for R1-Llama-8B on AIME, (b) ThinKV eviction curve. Impact of (c) refresh rate ($\tau$) for GPT-OSS-20B model on LiveCodeBench, (d) Impact of compression on generation length for R1-Llama-8B, (e) impact of block-size on throughput, (f) % breakdown of thoughts for R1-Llama-8B.

For a batch size of $B$, we issue $B$ parallel requests to emulate $B$ active users and measure the achieved system throughput (requests/s) together with the average end-to-end latency experienced by each user. The goal of this experiment is to evaluate performance when $B$ concurrent requests are actively being served. We report our findings in Figure 9 for R1-Llama-8B on A100-80GB GPU. We randomly sample $B$ AIME prompts and employ a cache budget of 1024 tokens. FullKV cannot sustain batch sizes beyond $B = 8$. Under an iso-batch comparison at $B = 8$, ThinKV achieves up to **58% lower latency** while sustaining higher request loads. At $B = 256$, again under iso-batch conditions, ThinKV achieves **38% higher reqs/s** and **27% lower latency** compared to R-KV.

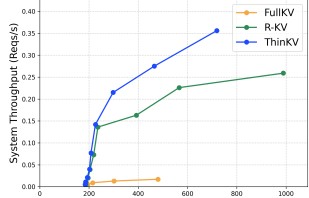

Figure 9: vLLM system throughput versus user-latency comparison.

## 6.3 DISCUSSIONS AND ABLATIONS

**Impact of ThinKV Components.** In Table 4, we ablate the accuracy, throughput, and latency contributions of ThinKV's components on GPT-OSS-20B using LiveCodeBench. For a fair comparison, we employ an iso-batch comparison with batch size of 8. TBQ, operating at an average precision of 3.5 bits, maintains accuracy comparable to FullKV. However, as shown in Figure 10(d), its substantial generation-length inflation negates most of the compression gains, yielding only a modest $1.1\times$ improvement in throughput.

TBE at smaller eviction budgets (e.g., 512) achieves large performance gains—up to $1.78\times$ higher throughput and $0.36\times$ lower latency—but at the cost of noticeable accuracy loss. At larger eviction budgets, TBE approaches near-lossless accuracy while still providing throughput improvements of up to $1.48\times$. ThinKV (TBQ+TBE) combines both mechanisms, delivering strong compression with only a marginal accuracy reduction. We would like to note that TBQ's

Table 4: Impact of ThinKV components on accuracy, performance (iso-batch) for GPT-OSS-20B on LiveCodeBench.

| Method | Avg. Precision / Eviction Budget | Accuracy | Batch Size | Norm. Throughput | Norm. Latency |
|---|---|---|---|---|---|
| FullKV | – | 77.8 | 8 | $1.0\times$ | $1.0\times$ |
| TBQ | 3.5 | 77.8 | 8 | $1.1\times$ | $0.98\times$ |
| TBE | 512 | 62.5 | 8 | $1.78\times$ | $0.36\times$ |
| TBE | 1024 | 76.9 | 8 | $1.48\times$ | $0.38\times$ |
| TBE | 2048 | 77.8 | 8 | $1.27\times$ | $0.44\times$ |
| **ThinKV (TBQ+TBE)** | **3.8, 1024** | **76.4** | **8** | **$1.51\times$** | **$0.42\times$** |

average precision is lower than ThinKV's because its inflated generation length introduces more transition tokens. Importantly, ThinKV achieves up to $1.51\times$ higher throughput and $0.42\times$ lower latency by avoiding the severe generation-length inflation exhibited by TBQ (see Figure 10(d)).

**Thought-Adaptive vs. Token-Level Heuristics.** To understand why ThinKV outperforms baselines, we analyze average recall rate of tokens with Top-10 attention scores (Tang et al., 2024) on R1-LLama-8B. Recall rate is the fraction of important tokens (Top-10) preserved by a compression method relative to those under full attention at each decoding step. As shown in Figure 10(a), ThinKV sustains recall rates close to FullKV across token budgets compared to R-KV and LazyEviction that rely on token-level heuristics that overlook reasoning structure.

**Compression Increases Generation Length.** In Figure 10(d), our R1-Llama-8B results show that pure quantization can inflate generation length by up to $5.1\times$. In contrast, eviction-based approaches do not induce such drastic inflation. ThinKV (TBQ+TBE) inherits this desirable stabilizing behavior and avoids the severe length expansion seen in quantization-only baselines.

**TBQ Precision.** In Figure 11(b), we study the effect of quantizing **R**, **T**, and **E** thoughts at different precisions for R1-Llama-8B on AIME and R1-Llama-70B on LiveCodeBench, using the notation R$x$E$y$T$z$ with $x, y, z \in \mathcal{B} = \{2, 4, 8\}$. We adopt R4E4T2 in all experiments due to its high accuracy and higher compression (also see §D.3).

**Eviction Behavior.** ThinKV's eviction strategy enforces proactive eviction (coarse-grained) in contrast to the fine-grained, stepwise eviction of H2O, R-KV. Figure 10(b) shows ThinKV's eviction behavior. As shown in Table 5, with ThinKV, the number of times a layer performs eviction across decode steps is minimal, 4.59% compared to R-KV's 82.93%, because R-KV waits for the budget to be exceeded to evict one token per decode step.

**Overhead Analysis.** In Table 5, we report operation-level breakdowns for R1-Llama-8B. The dequantization overhead of TBQ is included as part of the attention time. While TBE and thought refresh comprise $\sim 14\%$ of per-layer execution, their infrequent invocation ensures layers run overhead-free 95% of the time. Evidently, for R-KV, eviction and gather emerges as a major bottleneck (32.91%) since it is invoked in nearly every decoding step.

Table 5: Per-layer time breakdown (%) and call rates across decode steps.

| Operation | ThinKV | | R-KV | |
|---|---|---|---|---|
| | Time Breakdown (%) | # of Calls (%) | Time Breakdown (%) | # of Calls (%) |
| Thought Refresh | 3.80 | 0.7 | — | — |
| R-KV Eviction | — | — | 10.46 | 82.93 |
| Gather Time | 0 | 0 | 22.45 | 82.93 |
| TBE Eviction | 10.30 | 4.59 | — | — |
| Attention | 40.38 | 100 | 38.65 | 100 |
| MLP | 45.52 | 100 | 28.44 | 100 |

**Refresh Rate.** In Figure 10(c), we ablate different choices of refresh rate ($\tau$) for a GPT-OSS-20B model on LiveCodeBench. $\tau = 128$ offers the best trade-off between accuracy and overhead. Accuracy drops with larger $\tau$ as it skips thought changes and reduces opportunities to correct mispredictions.

**Optimal # of Layers.** In Figure 11(a), we ablate different $|\mathcal{L}^*|$ for R1-Llama-8B on LiveCodeBench. We select $|\mathcal{L}^*| = 4$ as it best balances accuracy and efficiency. Using all layers ($|\mathcal{L}^*| = 32$) degrades accuracy (§3.1).

**# of Thought Types.** In Figure 11(a), we show that $|\mathcal{T}| = 3$ yields the best accuracy on R1-Llama-8B evaluated on Live-CodeBench. For each $|\mathcal{T}|$, we select layers exhibiting $|\mathcal{T}|$ sparsity modes (can be less than $|\mathcal{L}^*|$) and quantize according to thought importance. When $|\mathcal{T}| < 3$, there is no notion of trajectory-changing thoughts. Therefore, eviction occurs only upon exceeding the KV budget (case 2 in Problem Formulation 2). See §E.10 for generalization to LLMs with $|\mathcal{T}| = 1$.

**Minimum Token Retention.** In Figure 11(a), we show why the minimum retention ($\mathcal{R}$) per thought segment is set to 4. Complete eviction ($\min \mathcal{R} = 0$) severely degrades accuracy, as the model loses track of reasoning trajectories and results in an endless reasoning loop. Retaining a minimal subset $\min \mathcal{R} = 4$ preserves semantic structure of reasoning and offers the best trade-off.

**% Breakdown of Thoughts.** Figure 10(f) shows the distribution of **R**, **T**, and **E** thoughts for R1-Llama-8B. Complex datasets (AIME) exhibit more transitions, than simpler ones (MATH-500).

**Block Size.** In Figure 10(e), we evaluate the effect of different block sizes on throughput. Block sizes of 8–16 deliver the best performance. Larger blocks, however, may pack more thought segments per block, incurring substantial metadata overhead in block table and increase eviction time.

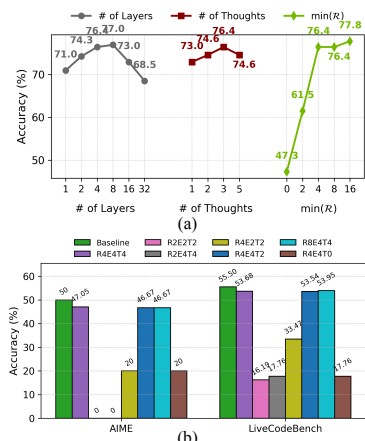

Figure 11: (a) Impact of $|\mathcal{L}^*|$, $|\mathcal{T}|$ and $\min \mathcal{R}$ on LiveCodeBench accuracy for R1-Llama-8B, (b) analysis of precision assignment for R1-Llama-8B on AIME and R1-Llama-70B on LiveCodeBench.

## 7 CONCLUSION

We introduced ThinKV, a thought-adaptive KV cache compression framework for LRMs. Exploiting attention sparsity, ThinKV decomposes chains of thought into reasoning, execution, and transition segments, enabling joint thought-aware quantization and eviction that sustains accuracy under high compression. On the system side, our Continuous Thinking kernel manages memory efficiently under dynamic decode-time eviction without costly compactions. This algorithm–system co-design delivers near-lossless accuracy with <5% of the original KV cache, while enabling up to $5.8\times$ throughput gains and substantially larger batch sizes across diverse reasoning benchmarks.

## ACKNOWLEDGEMENTS

We thank Reena Elangovan, Steve Dai, Yaosheng Fu, Zoey Song, Yingyan (Celine) Lin, Ben Keller, Andre Green, Souvik Kundu and Mingyu Lee for insightful discussions and valuable feedback that helped improve this work.

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

# APPENDIX

Table 6: Summary of notation used in the paper.

| Symbol | Description |
|---|---|
| $A$ | Final answer produced after reasoning |
| $L$ | Number of layers in the LRM |
| $y_i$ | Token generated at step $i$ |
| $Y_i$ | A thought segment consisting of multiple discrete tokens |
| $Y_0, \ldots, Y_{N-1}$ | Sequence of thought segments in a CoT output |
| $S_i^\ell$ | KV cache of layer $\ell$ after decoding step $i$ with associated thought type |
| $S_i^{\ell*}$ | Retained KV cache of layer $\ell$ after eviction |
| $(K_i^\ell, V_i^\ell)$ | Key and value vectors of token $y_i$ at layer $\ell$ |
| $\tilde{K}_i^\ell, \tilde{V}_i^\ell$ | Quantized key and value representations |
| $\mathcal{T} = \{c_0, \ldots, c_{T-1}\}$ | Set of $T$ thought categories |
| $\theta_1, \ldots, \theta_{T-1}$ | Sparsity thresholds separating thought categories |
| $\mathcal{L}^*$ | Optimal subset of layers |
| $\tau$ | Refresh interval for thought categorization |
| $\mathcal{B} = \{b_0, \ldots, b_{T-1}\}$ | Set of available quantization precisions |
| $\rho$ | Importance score function for thought categories |
| $\psi$ | Mapping from thought types to quantization precisions |
| $k$ | KV cache budget |
| $\pi$ | Eviction policy |
| $\mathcal{R}$ | Annealing schedule |

## A  OVERVIEW OF MATHEMATIC NOTATION

Table 6 summarizes the key notations used throughout the paper.

## B  EXTENDED RELATED WORKS

**Pre-LRM KV Cache Compression.** As LLMs began to support increasingly long contexts, the KV cache emerged as a primary target for optimization. Early work primarily addressed long input–context tasks by compressing the prefill KV cache. SnapKV (Li et al. (2024)), AdaKV (Feng et al. (2024)), and HeadKV (Fu et al. (2024)) prune tokens using attention statistics—via feature clustering or per-head budget allocation—while PyramidKV (Cai et al. (2024)) applies a pyramidal strategy, preserving more tokens in lower layers and compressing higher ones. These methods effectively reduce prompt memory but are ill-suited for LRMs, where the challenge lies in managing long outputs. To manage cache growth during decoding, methods such as StreamingLLM (Xiao et al. (2023)), ScissorHands (Liu et al. (2023)), H2O (Zhang et al. (2023)), MorphKV (Ghadia et al. (2025)), and KIVI (Liu et al. (2024b)) reduce memory through attention sinks, probabilistic retention, heavy-hitter selection, sliding windows, and uniform quantization, respectively. More recent works, including Q-Hitter (Zhang et al. (2024c)) and MiniKV (Sharma et al. (2025)), demonstrate that eviction and quantization can be co-designed, pointing toward hybrid strategies that maximize compression and throughput. While effective for extending traditional LLM outputs, these decode-time approaches often degrade accuracy on LRMs, as strategies driven by token recency or uniform compression fail to capture the reasoning progression and token importance characteristic of LRMs.

Compression approaches generally fall into four categories—eviction (Li et al. (2024); Ghadia et al. (2025); Zhang et al. (2023); Liu et al. (2023)), quantization (Liu et al. (2024b); Hooper et al. (2024)), merging (Nawrot et al. (2024); Wang et al. (2024); Liu et al. (2024a)), and low-rank decomposition (Kang et al. (2024); Sun et al. (2024)).

*Eviction:* StreamingLLM (Xiao et al. (2023)) retains a fixed-size sliding window together with a few attention sink tokens. MorphKV (Ghadia et al. (2025)) maintains a small set of recent tokens and selectively preserves older ones most correlated with the current context, providing constant-sized caches suitable for extended responses. LaCache (Shi et al. (2025)) introduces a ladder-shaped KV cache that preserves early tokens in shallow layers and later tokens in deeper layers, combined with iterative compaction of older caches, thereby supporting continuous long-context generation.

*Quantization:* Several works reduce KV cache memory by lowering precision while keeping all tokens. KVQuant (Hooper et al. (2024)) explores ultra-low precision by quantizing keys pre-RoPE,

applying sensitivity-aware non-uniform formats, and mixing dense/sparse quantization. More aggressive approaches investigate 1-bit quantization: methods such as Coupled Quantization (CQ) (Zhang et al. (2024b)) exploit inter-channel correlations to encode KV states with just 1 bit per channel, while calibration-based schemes (Han et al. (2025)) introduce scaling and correction factors to preserve accuracy.

*Merging:* Several works compress by consolidating semantically similar tokens. MiniCache (Liu et al. (2024a)) merges redundant prompt tokens into compact representations, while NACL (Chen et al. (2024)) prunes and merges tokens in a one-shot prefill step. These strategies reduce redundancy without per-step eviction but can blur token-level distinctions in reasoning tasks.

*Low-rank Decomposition:* Several works compress KV caches by factorizing them into low-rank representations to reduce memory and transfer costs. GEAR (Kang et al. (2024)) couples low-rank approximation with sparse correction to mitigate quantization errors. ShadowKV (Sun et al. (2024)) stores low-rank keys on the GPU while offloading values to CPU, reconstructing minimal sparse KV blocks on the fly. Other approaches such as LoRC (Zhang et al. (2024a)) and Palu (Chang et al. (2024)) apply progressive or layer-sensitive low-rank factorization of KV matrices, often in combination with quantization, to cut cache size and accelerate attention.

**Long Reasoning Compression.** A complementary line of work focuses on compressing the reasoning path rather than only the KV cache. Several approaches shorten chains-of-thought (CoT) at the output level: TALE (Han et al. (2024)) and SoT (Aytes et al. (2025)) guide models through prompt engineering to generate more concise explanations, while TokenSkip (Xia et al. (2025)) fine-tunes on condensed CoT datasets to reduce redundancy in multi-step reasoning. Other methods equip models with summarization capabilities, such as InftyThink (Yan et al. (2025)) and LightThinker (Zhang et al. (2025b)), which compress intermediate reasoning into summaries to save tokens. A different direction operates in latent space, with approaches like CCoT (Cheng & Van Durme (2024)) and SoftCoT (Xu et al. (2025)) enabling reasoning directly on compressed internal representations rather than verbose token sequences. Most recently, RPC (Song et al. (2025)) adaptively prunes, merges, or reorders reasoning trajectories while preserving correctness.

**System-Level Optimizations.** System-level methods complement algorithmic compression by managing KV storage at runtime. Quest (Tang et al. (2024)) loads only query-relevant KV pages, while OmniKV (Hao et al. (2025)) streams KV from CPU in small chunks to reduce GPU memory pressure—though both retain $O(N)$ complexity in sequence length $N$. MiniKV (Sharma et al. (2025)) introduces FlashAttention-compatible kernels for compressed KV, and Q-Hitter (Zhang et al. (2024c)) unifies eviction and quantization to reduce GPU I/O overhead. H2O (Zhang et al. (2023)) and KVZip (Kim et al. (2025)) avoid costly gather operations with ring-buffered caches, while MemShare (Chen et al. (2025a)) enables block-level KV reuse across reasoning segments.

## C  SUPPLEMENTARY BACKGROUND

### C.1  LRM INFERENCE STAGES

The inference process of an $L$-layer LRM proceeds in two distinct phases: the *prefill stage*, which processes the input prompt, and the *decode stage*, which generates the output autoregressively. These phases differ fundamentally in their parallelism and computational bottlenecks.

**Prefill.** Given a prompt of length $l_{\text{prompt}}$, the model embeds the input into hidden representations $X \in \mathbb{R}^{b \times l_{\text{prompt}} \times d}$, where $b$ is the batch size and $d$ the hidden dimension. For each layer $\ell$, keys and values are computed as

$$X_K = XW_K^\ell, \quad X_V = XW_V^\ell,$$

with $W_K^\ell, W_V^\ell \in \mathbb{R}^{d \times d}$ denoting the projection matrices. The resulting KV tensors $\{(K_j^\ell, V_j^\ell)\}_{j=0}^{l_{\text{prompt}}-1}$ are stored in $S_{l_{\text{prompt}}}^\ell$ for subsequent use. Since all prompt tokens are processed in parallel, the prefill stage is dominated by quadratic attention cost in $l_{\text{prompt}}$ and is typically *latency-bound*.

**Decode.** Once the cache has been initialized, generation proceeds autoregressively. At decode step $i$, the current token embedding $y_i$ produces

$$K_i^\ell = y_iW_K^\ell, \quad V_i^\ell = y_iW_V^\ell, \quad q_i^\ell = y_iW_Q^\ell,$$

which are appended to the existing cache:

$$S_i^\ell \leftarrow S_{i-1}^\ell \cup \{(K_i^\ell, V_i^\ell)\}.$$

Attention is then computed against all cached keys:

$$A_i^\ell = \text{softmax}\left(\frac{q_i^\ell (K_{0:i}^\ell)^\top}{\sqrt{d}}\right), \qquad O_i^\ell = A_i^\ell V_{0:i}^\ell.$$

This process repeats for $l_{\text{gen}}$ output tokens. Unlike prefill, decoding reuses the cache and extends it one token at a time, making the stage inherently *throughput-bound* due to repeated KV cache lookups and memory traffic.

In summary, prefill amortizes computation across the entire prompt to initialize the cache, while decode iteratively expands the cache to produce the final output sequence.

## C.2 ATTENTION MECHANISMS

We briefly summarize two widely adopted attention variants: *Multi-Head Attention (MHA)* and *Grouped-Query Attention (GQA)*. ThinKV is applicable to both attention variants.

**Multi-Head Attention (MHA).** In the autoregressive setting, each decode step produces a single query vector $q_h \in \mathbb{R}^{1 \times d}$ for head $h$, which attends over the stored key vectors $K_h \in \mathbb{R}^{n \times d}$ from the $n$ past tokens. The attention matrix is given by,

$$a_h = \text{softmax}\left(\frac{q_h K_h^\top}{\sqrt{d}}\right) \in \mathbb{R}^{1 \times n}. \tag{1}$$

The attention weights are then applied to the value states $V_h \in \mathbb{R}^{n \times d}$, and the outputs from all heads are concatenated and projected back to the hidden dimension. For sparsity analysis, attention scores are averaged across all heads.

**Grouped-Query Attention (GQA).** In GQA, several query heads share a common set of key and value states. For a head group indexed by $h$, the cached keys and values are $(K_h, V_h) \in \mathbb{R}^{n \times d}$, while $G$ distinct query vectors $\{q_{h,g}\}_{g=0}^{G-1}$ are produced within the group. The attention score for query head $g$ is given by

$$a_{h,g} = \frac{q_{h,g} K_h^\top}{\sqrt{d}} \in \mathbb{R}^{1 \times n}. \tag{2}$$

These per-query matrices are aggregated element-wise across the group using max pooling:

$$a_h^{\text{group}} = \text{maxpool}\left(a_{h,0}, \ldots, a_{h,G-1}\right) \in \mathbb{R}^{1 \times n}. \tag{3}$$

Finally, the consolidated scores are renormalized along the key dimension to obtain the final attention weight $a_h$ for the group,

$$a_h = \text{softmax}\left(a_h^{\text{group}}\right) \in \mathbb{R}^{1 \times n}. \tag{4}$$

For sparsity analysis, attention scores are averaged across groups.

## C.3 KV PERMUTATION INVARIANCE OF ATTENTION

**Theorem 1** (KV Permutation Invariance of Attention). *Given* $q \in \mathbb{R}^{1 \times d}$, $K \in \mathbb{R}^{n \times d}$, $V \in \mathbb{R}^{n \times d}$, *define*

$$o = \text{softmax}\left(\frac{qK^\top}{\sqrt{d}}\right) V \in \mathbb{R}^{1 \times d}.$$

*For any permutation matrix* $\Pi \in \mathbb{R}^{n \times n}$,

$$\text{softmax}\left(\frac{q(\Pi K)^\top}{\sqrt{d}}\right)(\Pi V) = \text{softmax}\left(\frac{qK^\top}{\sqrt{d}}\right) V.$$

*Proof.* Let $s = \frac{1}{\sqrt{d}} qK^\top \in \mathbb{R}^{1 \times n}$. Since $\Pi$ is a permutation matrix, $\Pi^\top \Pi = I$, and for any $u \in \mathbb{R}^{1 \times n}$ we have

$$\text{softmax}(u\Pi^\top) = \text{softmax}(u)\Pi^\top \quad \textit{(Equivariance Property)}$$

---

**Algorithm 1:** Calibration Process for Thought Decomposition

---

1: **Input:** Pre-trained LRM $\mathcal{M}$ with $L$ layers, calibration dataset $\mathcal{D}$ of $P$ prompts, number of thought types $T$, optimal number of layers $\ell^*$
2: **Output:** Optimal layer subset $\mathcal{L}^*$, sparsity threshold set $\Theta = \{\theta_1, \ldots, \theta_{|\mathcal{T}|-1}\}$
3: Initialize $\mathcal{U}_\ell$ for each layer $\ell$
4: **for** each prompt $p \in \mathcal{D}$ **do**
5:     Run $\mathcal{M}$ on $p$ and generate sequence of length $M_p$
6:     **for** each decoding step $t \in [M_p]$ **do**
7:         **for** each layer $\ell \in [L]$ **do**
8:             Compute sparsity $u$ from attention scores
9:             Append $u$ to $\mathcal{U}_\ell[p][t]$
10:         **end for**
11:     **end for**
12: **end for**
13: Initialize $\mathcal{L}^* \leftarrow \emptyset$
14: **for** each prompt $p$ **do**
15:     Initialize $L^*[p] \leftarrow \emptyset$
16:     **for** each layer $\ell$ **do**
17:         Apply KDE $\hat{f}_h(x) = \frac{1}{Mh} \sum_{m=1}^{M} K\left(\frac{x - x_m}{h}\right)$ on $\mathcal{U}_\ell[p]$
18:         Estimate modes $\Omega_\ell^{(p)} = \{x \mid \hat{f}_h'(x) = 0, \hat{f}_h''(x) < 0\}$
19:         **if** $|\Omega_\ell| = T$ **then**
20:             Add $\ell$ to $L^*[p]$
21:         **end if**
22:     **end for**
23: **end for**
24: $\mathcal{L}^* \leftarrow \bigcap_{p=1}^{P} L^*[p]$
25: **for** each layer $\ell \in \mathcal{L}^*$ **do**
26:     **for** each prompt $p \in [P]$ **do**
27:         Identify local minima of the KDE and record thresholds $\{\theta_1^{(\ell,p)}, \ldots, \theta_{|\mathcal{T}|-1}^{(\ell,p)}\}$
28:     **end for**
29: **end for**
30: Compute final thresholds $\theta_j = \frac{1}{|\mathcal{L}^*|P} \sum_{\ell \in \mathcal{L}^*} \sum_{p=1}^{P} \theta_j^{(\ell,p)} \quad \forall j \in [|\mathcal{T}| - 1]$
31: **return** $\mathcal{L}^*, \{\theta_1, \ldots, \theta_{|\mathcal{T}|-1}\}$

---

Applying this with $u = s$ yields

$$\text{softmax}\left(\tfrac{1}{\sqrt{d}} q(\Pi K)^\top\right)(\Pi V) = \text{softmax}(s\Pi^\top)(\Pi V)$$
$$= (\text{softmax}(s)\Pi^\top)(\Pi V)$$
$$= \text{softmax}(s)(\Pi^\top \Pi)V$$
$$= \text{softmax}(s)V$$

$\square$

*Remark.* The same invariance holds for GQA: for any group $h$ with shared $(K_h, V_h)$, a joint permutation of their rows leaves the group attention output unchanged.

*Remark.* This permutation invariance explains why ThinKV can avoid reordering the KV cache during attention computation.

## C.4 GROUP QUANTIZATION

Group quantization Ramachandran et al. (2025c) reduces precision by partitioning tensors into fixed-size groups and sharing a scale (and optionally zero-point) within each group. Given a tensor $X \in \mathbb{R}^{n \times d}$ and group size $g$, the entries are divided into groups $X_{G_i}$ of length $g$. Each group is quantized

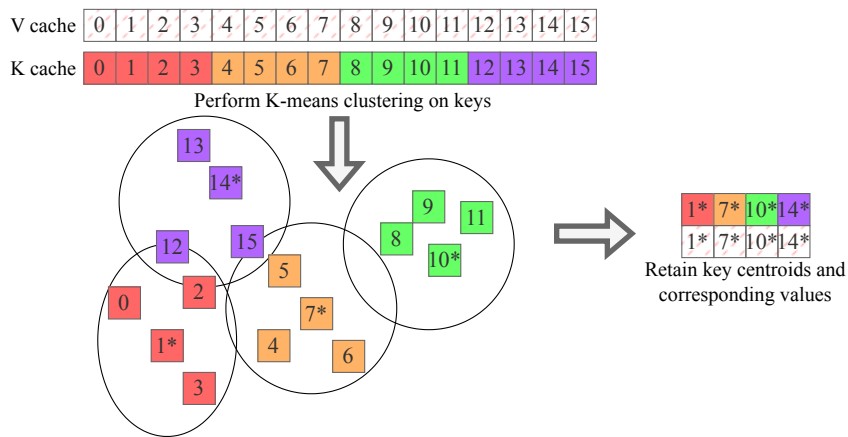

Figure 12: Illustration of eviction policy $\pi$'s k-means-based eviction mechanism.

Table 7: Keyword list to interpret different thought types.

| | |
|---|---|
| **Reasoning** | Think, Approach, Remember, Find, Okay, Suppose, Verify |
| **Transition** | Wait, Hmm, Wait no, Alternatively, But wait, Earlier I said that |
| **Execution** | Now, The steps are, Mathematical equations, Code syntax |

as

$$\hat{X}_{G_i} = \text{round}\left(\frac{X_{G_i}}{\Delta_i}\right), \qquad \Delta_i = \frac{\max(X_{G_i})}{2^b - 1},$$

where $b$ is the target bit-width and $\Delta_i$ is the group-specific scale.

Smaller group sizes yield tighter ranges and lower error, while larger groups reduce metadata overhead. Group quantization thus provides a flexible trade-off between accuracy and efficiency, and serves as the default scheme for low-bit KV cache quantization in LRMs.

### C.5 PAGED ATTENTION

PagedAttention is an attention algorithm introduced in vLLM to address the inefficiencies of managing key–value (KV) cache memory during large language model serving. Traditional systems store each request's KV cache in contiguous memory, leading to severe internal and external fragmentation as output lengths vary, and preventing memory sharing across sequences. Inspired by virtual memory paging, PagedAttention partitions the KV cache into fixed-size blocks that can be stored non-contiguously in GPU memory. Logical blocks are dynamically mapped to physical blocks through block tables.

## D SUPPLEMENTARY DETAILS ON THINKV

### D.1 THOUGHT DECOMPOSITION CALIBRATION PROCESS

Algorithm 1 depicts the algorithm for the offline calibration stage. This process estimates the sparsity thresholds that separate different thought categories by analyzing layer-wise attention sparsity distributions over a calibration dataset.

### D.2 THOUGHT KEYWORD LIST

To aid interpretation of sparsity regions, we provide representative keywords for the three thought types in Table 7. These keywords are illustrative and only serve to map sparsity regions to reasoning, execution, and transition thoughts. They are not used for thought identification during inference.

### D.3 QUANTIZATION DATA FORMATS

We employ three element formats of different precision levels:

**FP8 (E4M3).** This is an 8-bit floating-point format with 1 sign bit, 4 exponent bits, and 3 mantissa bits. It provides a balance between dynamic range and accuracy and serves as the highest-precision option for thought-adaptive quantization, used primarily for reasoning tokens. This format only uses a per-tensor FP32 scale factor.

**NVFP4.** NVIDIA's recently introduced 4-bit floating-point format, NVFP4 (Alvarez et al., 2025), combines 1 sign bit, 2 exponent bits, and 1 mantissa bit optimized for inference workloads. NVFP4 employs a group-wise scale factor (Ramachandran et al., 2025a) with FP8 (E4M3) representation and a group size of 16. Execution and reasoning tokens are stored in NVFP4 to reduce memory footprint while retaining sufficient accuracy.

**Ternary (2-bit).** This format encodes each element with two bits, covering three distinct values $\{-1, 0, +1\}$. Of the four possible codes, one corresponds to $-0$, which is redundant and simply mapped to $0$.Similar to above, ternary also employs a group-wise scale factor with FP8 (E4M3) representation and a group size of 16. In our design, ternary quantization is applied exclusively to transition thoughts, where lower precision can be tolerated with minimal impact on overall accuracy.

Together, these formats enable a precision hierarchy (FP8 > NVFP4 > Ternary) aligned with the observed importance of reasoning, execution, and transition thoughts.

### D.4 TBE EVICTION POLICY

Figure 12 illustrates the K-means eviction process. When a thought segment is selected for eviction, we cluster the post-RoPE key embeddings into a target number of groups, determined by the annealing schedule $\mathcal{R}$. Each cluster is replaced by its centroid key, and the corresponding value entry is retained. As shown, color-coded blocks indicate tokens that are close in the embedding space; centroids (marked with a star) are selected from each cluster, and only these representative key–value pairs are preserved in the cache.

While prior work (Hooper et al., 2025) has highlighted that RoPE can induce token drift, thereby complicating the clustering of keys, we observe that this effect is negligible when clustering is restricted to tokens within a single thought segment. Each thought segment spans only 128 tokens, and the limited span ensures that RoPE-induced drift remains minimal, in contrast to clustering over the entire chain of thought (CoT) as done in (Hooper et al., 2025), where the drift accumulates more substantially. Furthermore, if future evidence suggests that drift becomes noticeable even within a thought segment, the Windowed RoPE strategy (He et al., 2025) can be readily employed as a complementary technique to mitigate this issue.

### D.5 THINKV PSEUDOCODE

```
def generation_loop(prompt, max_gen_len, L, params):
    # Prologue
    init_block_tables()
    init_kv_cache()
    thresholds = (theta_low, theta_high)
    refresh_period = params.refresh
    group_size = params.group_size
    budget = params.token_budget

    # Generate
    for i in range(max_gen_len):
        for l in range(L):
            # Forward attention
            q, k_fp, v_fp = project_qkv(h[l])

            # Thought refresh: 0=transition, 1=execution, 2=reasoning
            if i % refresh_period == 0:
                spars = measure_sparsity(l)
```

```
                prev_thought[l] = thought[l]
                thought[l] = classify(spars, thresholds)

            # TBQ: group quantization
            buffer_add(l, k_fp, v_fp)
            if buffer_size(l) >= group_size:
                k_grp, v_grp = buffer_take(l, group_size)
                if thought[l] == 2:
                    kq, vq = Q4(k_grp, v_grp)    # NVFP4
                elif thought[l] == 1:
                    kq, vq = Q4(k_grp, v_grp)    # NVFP4
                else:
                    kq, vq = Q2(k_grp, v_grp)    # ternary
                kv_cache_update(l, kq, vq)

            # TBE: anneal at end of each transition segment
            if i % refresh_period == 0 and prev_thought[l] == 0:
                prev_segments = find_segments_before(l, step=i)
                for seg in prev_segments:
                    t = seg.type
                    keep = anneal_size(t)
                    ids = kmeans_select(l, seg, keep)
                    mark_evicted(l, seg, ids)

            # TBE: budget-constrained eviction
            if kv_size(l) > budget:
                candidates = active_thought_types(l)
                t = argmin_importance(candidates)
                oldest = find_oldest_segment(l, t)
                keep = anneal_size(t)
                ids = kmeans_select(l, oldest, keep)
                mark_evicted(l, oldest, ids)

            # Attention computation
            h[l+1] = attend(q, K[l], V[l])

    # Epilogue
    return decode_tokens()
```

Listing 1: ThinKV generation loop.

### D.6 THINKV WALKTHROUGH EXAMPLE

We provide a detailed walkthrough of ThinKV using the illustration in Figure 6.

**TBQ Quantization.** During decoding, tokens are first appended to $B_{\text{buf}}$ in full precision. Once the group size is reached, they undergo group quantization. In the illustration, we highlight reasoning (**R**) tokens, which are quantized into the NVFP4 format. It is important to note that the block table indexes only quantized tokens i.e., the block table updates at group-size granularity.

**Step a.** Following quantization, CT kernel queries the block table to determine whether a physical block of type-2 (reasoning) tokens has available capacity. Since the table is initially empty, a new entry is created with thought type 2, and a physical block is allocated. The start index of this reasoning segment is recorded as 0. Because the block currently stores only a single segment, the segment mask is initialized to all 1s, while the eviction mask remains all 0s.

**Step b.** When token 'D' is generated, a refresh occurs, switching to a type-1 (execution) thought. Execution tokens are likewise group quantized to NVFP4. CT then allocates a new entry for the execution thought type. Importantly, CT enforces thought-aware paging: execution tokens are never placed into partially filled blocks of other thoughts, even if capacity remains.

**Step c.** Beginning with token 'I', the decode refreshes to type-0 (transition) tokens. As defined in §4.3, the end of this transition segment (the 'L' token) triggers the TBE kernel. The kernel scans the block table, identifies all prior segments via their start indices, and applies the eviction pol-

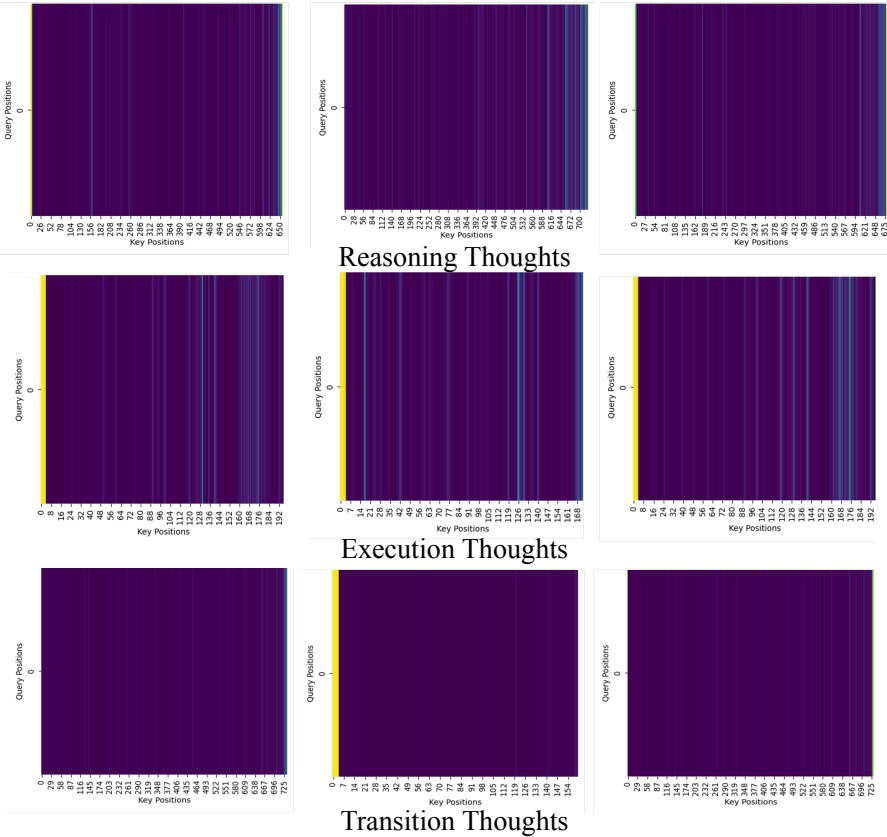

Figure 13: Visualization of attention maps across different thought types. At decode time only a single query is present; maps are broadcasted for clarity of visualization.

icy. Instead of physically removing tokens, the eviction mask is updated to mark evicted positions, deferring eviction.

**Step d.** After the next refresh, decoding returns to reasoning. CT inspects the eviction mask to identify available slots in existing reasoning blocks. For tokens 'M' and 'N', it locates two free slots in physical block 4, places the tokens there, and resets the eviction mask to all 0s once the slots are filled. In parallel, it appends the start index of the new reasoning segment and updates the segment mask to indicate the token positions for each segment. By reusing evicted slots in this way, ThinKV achieves efficient memory utilization without introducing additional HBM bandwidth pressure or stalling the inference critical path. For tokens 'O' and 'P' since there are no empty slots available, a new block is allocated.

# E  EXTENDED EVALUATIONS

## E.1  DATASET DETAILS: AIME

Following Cai et al. (2025); Liu et al. (2025), we construct an AIME benchmark of 30 prompts, comprising 15 prompts sampled from AIME 2024 and 15 from AIME 2025.

## E.2  EVALUATION SETUP DETAILS

We use the latest model checkpoints available on Hugging Face for all evaluations. We build on the Hugging Face Transformers codebase and implement the ThinKV algorithm by modifying it. The Hugging Face Transformers codebase employs the FlashAttention-2 kernel as its default attention backend, which we leverage for all baseline comparisons. In addition, we modify a Triton

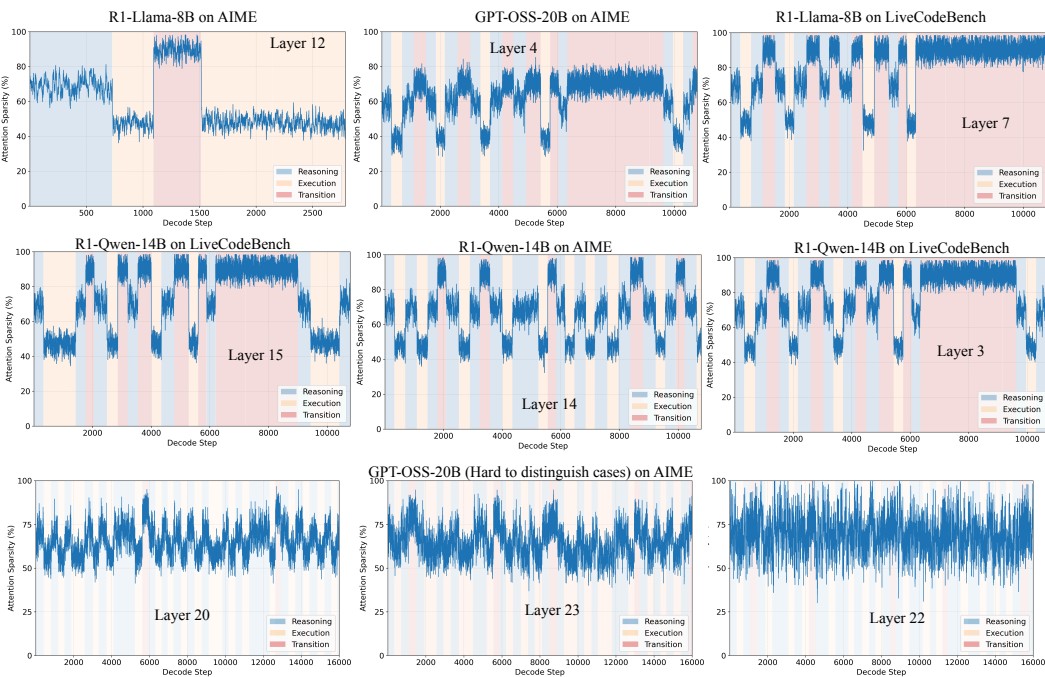

Figure 14: Layer-wise attention sparsity across decode steps for different models and datasets.

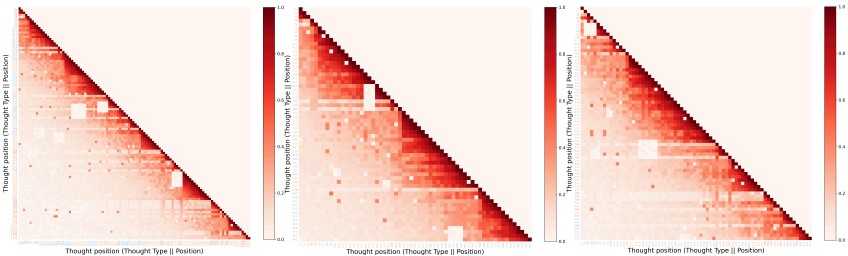

Figure 15: Additional visualization of pairwise thought associations for different input prompts from different datasets (AIME and LiveCodeBench).

implementation of PagedAttention and integrate it into the Hugging Face Transformers stack; this baseline PagedAttention supports all features present in vLLM's implementation. This integration was carried out as a proof of concept to quickly evaluate ThinKV's performance. This proof-of-concept serves as a stepping stone toward full integration with optimized inference engines. Although this stack is not the most optimized, we still expect commensurate improvements when running on frameworks such as vLLM, as ThinKV's modifications are orthogonal to specific kernel implementations. To validate this, we integrate ThinKV inspired by this PR in vLLM vLLM PR 16160 (2025). Our integration targets only the vLLM v1 version. Specifically, our major modifications are centered around 'block_table.py', 'flash_attn.py' and 'csrc/attention'. By adjusting the flags in 'envs.py', we can seamlessly toggle between R-KV, ThinKV, and a no-compression (Full-KV) baseline, enabling comparisons within the same vLLM framework.

For measuring gather overhead, we profile this behavior on A100 and H200 GPUs using NVIDIA Nsight (Nsight, 2025).

### E.3 VISUALIZATION OF ATTENTION MAPS

Figure 13 shows the attention weight matrices at different decoding steps, each corresponding to a single query. The visualization reveals that transition thoughts exhibit the highest sparsity, followed by reasoning, and then execution.

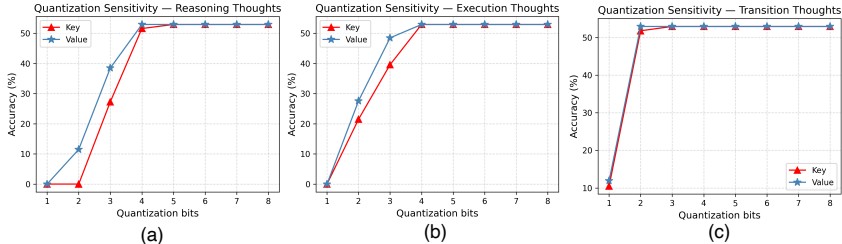

Figure 16: Quantization sensitivity analysis of KV cache for (a) reasoning, (b) execution and (c) transition thoughts.

### E.4    ATTENTION SPARSITY PLOTS

In Figure 14, we present attention sparsity across decode steps for several model families. For GPT-OSS-20B in particular, we highlight layers where the sparsity structure is difficult to distinguish, leading to ambiguous or poorly defined boundaries between thought categories.

### E.5    PAIRWISE THOUGHT ASSOCIATION MAPS

In Figure 15, we show the inter-thought dynamics for additional prompts drawn from AIME and LiveCodeBench.

### E.6    RESULTS ON MOBILELLM-R1 950M (GSM8K)

For GSM8K, we set the KV cache budget to 256 tokens for an average generation length of ∼1500. Under this setting, ThinKV operates at an average precision of 3.9 bits and achieves a $24\times$ compression ratio while maintaining accuracy comparable to R-KV, which compresses at only $6\times$. This demonstrates ThinKV's effectiveness in sustaining reasoning quality under high compression even for lightweight models such as MobileLLM-R1 950M.

Table 8: Comparison of ThinKV and R-KV on GSM8K using MobileLLM-R1-950M.

| Method | Compression | GSM8K |
|--------|-------------|-------|
| FullKV | 1 | 67.5 |
| R-KV | 6 | 60.8 |
| **ThinKV** | **24** | **60.1** |

### E.7    RESULTS ON GPT-OSS 120B (LIVECODEBENCH)

We evaluate ThinKV on GPT-OSS 120B using Live-CodeBench under a fixed KV budget of $k = 1024$ tokens. GPT-OSS exposes a *reasoning effort* knob (low/medium/high) that controls the model's reasoning budget; we sweep medium and high settings in our study. Across both effort levels, ThinKV tracks FullKV closely: at *high* effort, ThinKV attains 67.5 vs. 69.4 for FullKV ($-1.9$ points); at *medium*, 59.3 vs. 61.8 ($-2.5$ points). Higher effort predictably yields better accuracy but longer generations, increasing KV stress; ThinKV sustains accuracy under this regime despite the the 1024-token cache. Across both reasoning efforts ThinKV maintains an average precision of 3.6-bits.

Table 9: Accuracy of ThinKV vs FullKV across reasoning effort levels for GPT-OSS-120B on Live-CodeBench.

| Method | Reasoning Effort | Accuracy |
|--------|------------------|----------|
| FullKV | High | 69.4 |
| **ThinKV** | **High** | **67.5** |
| FullKV | Medium | 61.8 |
| **ThinKV** | **Medium** | **59.3** |

### E.8    ABLATION ON DATA FORMATS

We further investigate the impact of different data formats on ThinKV. Specifically, we ablate the use of conventional integer quantization, where we employ INT4 and INT2 representations with same scaling as described in §D.3. This allows us to isolate the effect of the number representation from the scaling strategy. As shown in Table 10, ThinKV with INT4/INT2 suffers notable accuracy degradation on both AIME and LiveCodeBench. This demonstrates the combination of NVFP4 and ternary data format as the better choice.

Table 10: Impact of data format choices on accuracy for R1-Llama-8B.

| Method | AIME | LiveCodeBench |
|--------|------|---------------|
| Baseline | 50 | 32.14 |
| ThinKV w/ INT | 46.7 | 28.5 |
| **ThinKV** | **50** | **32.14** |

### E.9 Quantization Sensitivity Analysis

Following Cheng et al. (2025), we analyze the quantization sensitivity of the KV cache across reasoning, execution, and transition thoughts in Figure 16. Using INT quantization on R1-Llama-70B (LiveCodeBench), we sweep the precision of either K or V within a single thought type while fixing all remaining KV entries to 8-bit. The results show that transition thoughts are highly robust—both K and V tolerate aggressive quantization—supporting our use of 2-bit precision. Execution thoughts similarly remain stable down to 4 bits. In contrast, the K cache of Reasoning thoughts is significantly more sensitive, consistent with the K/V asymmetry observed in Cheng et al. (2025), while the corresponding V cache remains resilient. These findings directly validate the precision assignments adopted in ThinKV.

### E.10 Generalization to LLMs

To evaluate ThinKV's generalizability beyond LRMs, we test it on the long-response benchmark LongWriter (Bai et al., 2024), which includes 60 prompts across domains such as emails, blogs, essays, and novels, with response lengths ranging from 100 to 12K words. Following Zhang et al. (2023), we constrain the KV cache budget to $5\%$ of decode tokens.

Table 11: LLM accuracy comparison on LongWriter task.

| Method | Llama-8B | Phi-14B |
|---|---|---|
| FullKV | 66.5 | 62.9 |
| H2O (5%) | 68.1 | 61.5 |
| **ThinKV (3.75%)** | **67.9** | **63.8** |

Unlike LRMs, LLMs do not exhibit distinct thought types; hence, we set $|\mathcal{T}| = 1$ with $\mathcal{B} = 4$, treating all tokens as a single category. In this setting, eviction occurs only when the cache budget is reached, after which prior tokens are annealed to the nearest power of two. For evaluation, we follow Ghadia et al. (2025) and use an LLM-based judge (Mistral-Large-123B) to score responses across multiple criteria. As shown in Table 11, ThinKV generalizes effectively to LLMs, matching or even surpassing H2O while delivering higher compression through its hybrid scheme.

### E.11 Pareto-front Analysis

Figure 17 illustrates the relationship between KV-cache size and accuracy across several SoTA compression and eviction baselines for R1-Llama-70B on LiveCodeBench. For this analysis, inspired by (Sharma et al., 2025), we sweep different configurations (token budget, quantization precision) for each of the evaluated methods. Methods such as LazyEviction, PM-KVQ, and R-KV achieve moderate compression but suffer significant accuracy degradation, while high-accuracy configurations require substantially larger KV budgets. In contrast, ThinKV consistently delivers near–FullKV accuracy at dramatically smaller KV-cache sizes, tracing a dominant curve that establishes the new Pareto frontier. Specifically, most ThinKV configurations lie strictly above competing methods at equivalent or smaller memory footprints. This frontier shift highlights ThinKV's ability to achieve the best possible trade-

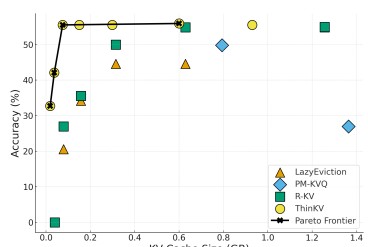

Figure 17: Accuracy vs KV cache size comparison of ThinKV against SoTA baselines for R1-Llama-70B on LiveCodeBench.

off between accuracy and memory, outperforming both quantization-only and eviction-only approaches and confirming its strong scalability across compression regimes.

### E.12 Throughput Evaluation of ThinKV in vLLM

As shown in Table 12, we report throughput under two iso-batch comparisons: (i) batch size = 8 against FullKV and R-KV (ovl), and (ii) batch size = 256 against R-KV (ovl). All methods have been implemented in vLLM for a fair comparison and measurements conducted on an A100-80GB GPU. At a batch size of 8, ThinKV delivers higher throughput than both FullKV and R-KV (ovl), improving over FullKV by more than 50%. At a larger batch

Table 12: Throughput comparison under different batch sizes implemented in vLLM.

| Method | Batch Size | Budget | Throughput |
|---|---|---|---|
| FullKV | 8 | – | 228.5 |
| R-KV (ovl) | 8 | 1024 | 331.9 |
| **ThinKV** | **8** | **1024** | **346.9** |
| R-KV (ovl) | 256 | 1024 | 4883.3 |
| **ThinKV** | **256** | **1024** | **6622.4** |

size of 256, ThinKV's advantage becomes more pronounced: it achieves a substantial throughput

increase over R-KV (ovl) of up to $1.35\times$. ThinKV demonstrates superior scalability by eliminating gather-based compaction and achieving higher KV-cache compression, both of which translate directly into faster model execution.

## E.13 EXPERIMENTS ON QWEN3 MODELS

The Qwen3 model family (Yang et al., 2025) enables seamless switching between thinking and non-thinking modes via flags. Using a representative Qwen3-8B model, we compare its non-thinking mode against ThinKV-enabled thinking mode. ThinKV achieves $< 2.2\%$ accuracy drop across eviction budgets while using $< 6.87\%$ of FullKV memory. In contrast, the non-thinking mode exhibits a drastic $> 33\%$ accuracy degradation. This highlights that reasoning-augmented decoding is essential for correctness.

Table 13: Accuracy comparison between thinking, non-thinking, and ThinKV-enabled thinking modes on Qwen3-8B evaluated on Live-CodeBench.

| Method | Mode | Avg. Precision / Eviction Budget | Accuracy (%) |
|---|---|---|---|
| FullKV | Non-Thinking | – | 21.8 |
| FullKV | Thinking | – | 55.6 |
| **ThinKV** | **Thinking** | **3.6 / 1024** | **53.4** |
| **ThinKV** | **Thinking** | **3.7 / 2048** | **55.2** |

## E.14 LATENCY BREAKDOWN ACROSS BATCH SIZES

This experiment is conducted to better understand how the performance of ThinKV's components scale across batch sizes. For this analysis, we focus on a representative decode step that includes all mechanisms in action. Figure 18 measurements show that ThinKV's overhead (TBE eviction + thought refresh) remains minimal across batch sizes, consistently accounting for only ∼14% of the total latency, while Attention and MLP operations dominate with more than 80–85% of the runtime. As batch size increases, the proportion of time spent in core model execution (attention, MLP) grows, confirming that ThinKV scales efficiently with increasing batch size.

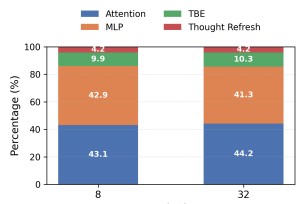

Figure 18: Latency breakdown across different batch sizes.

## E.15 TIME-PER-REQUEST ANALYSIS

Table 14 reports the average end-to-end request latency (Time-per-Request, TPR), accuracy, and Intelligence/Watt (Saad-Falcon et al., 2025) for various KV-compression strategies evaluated on the AIME benchmark using R1-Llama-8B. ThinKV at a token budget of 1024, while simultaneously achieving lossless compression, is able to achieve up to 6% lower latency on average per request as compared to the FullKV baseline. These gains extend beyond what a highly optimized framework like vLLM already provides, and ThinKV's benefits become es-

Table 14: Comparison of Time-per-Request (TPR), Accuracy, and Intelligence/Watt (Intel./Watt).

| Method | Token Budget | TPR (s) | Accuracy (%) | Intel./Watt |
|---|---|---|---|---|
| FullKV | – | 259.6 | 50.0 | 0.20 |
| R-KV (seq) | 512 | 242.6 | 40.0 | 0.17 |
| R-KV (ovl) | 512 | 240.8 | 40.0 | 0.17 |
| **ThinKV** | **512** | **237.5** | **46.7** | **0.21** |
| R-KV (seq) | 1024 | 247.8 | 46.7 | 0.20 |
| R-KV (ovl) | 1024 | 246.0 | 46.7 | 0.20 |
| **ThinKV** | **1024** | **243.6** | **50.0** | **0.22** |
| R-KV (seq) | 2048 | 254.2 | 50.0 | 0.20 |
| R-KV (ovl) | 2048 | 253.7 | 50.0 | 0.20 |
| **ThinKV** | **2048** | **251.0** | **50.0** | **0.21** |

pecially pronounced at larger batch sizes. Recent works have demonstrated that Intelligence/Watt (Saad-Falcon et al., 2025) offers a unified view of both capability and efficiency, making it a principled metric for comparing compression strategies. As shown in Table 14, these latency improvements materially increase ThinKV's Intelligence/Watt over FullKV and R-KV.

## E.16 INTEGRATION WITH SNAPKV

ThinKV is orthogonal to other prefill-optimization techniques, and can be combined with them seamlessly. To illustrate this, we evaluate ThinKV integrated with SnapKV on LongBench v2 **?**. LongBench v2 is a suite of long-context evaluation tasks characterized by both large input contexts and substantial output lengths. For this study, we evaluate on R1-LLama-8B and select the Code Repo QA

Table 15: Ablation of Prefill and Decode Settings for Hybrid (SnapKV + ThinKV) compression.

| Method | Prefill Precision / Eviction Budget | Decode Precision / Eviction Budget | Accuracy |
|---|---|---|---|
| FullKV | – | – | 30 |
| **ThinKV** | | **3.8 / 512** | 28 |
| **SnapKV + ThinKV** | **4-bits / 2048** | **3.7 / 512** | 24 |
| **SnapKV + ThinKV** | **4-bits / 4096** | **3.8 / 512** | 28 |

task from LongBench v2, which features 16K–32K average input lengths and output sequences up

to 8K tokens, making it a representative long-input/long-output setting. Prefill tokens are quantized to 4 bits using TBQ and evicted according to SnapKV's eviction policy under varying budgets, while the decode stage employs ThinKV with a fixed KV-cache budget of 512 tokens. Our results are summarized in Table 15. ThinKV incurs only a small accuracy drop, despite achieving $4\times$ prefill compression and up to $66\times$ decode compression. When combined with SnapKV, a 4096 prefill token budget results in only $2\%$ accuracy degradation.

### E.17 LRM EXAMPLE REASONING TRACE

---

**Input Prompt 1:** Let $ABCD$ be a rhombus whose vertices all lie on the hyperbola $\frac{x^2}{20} - \frac{y^2}{24} = 1$ and are in that order. If its diagonals intersect at the origin, find the largest number less than $BD^2$ for all rhombuses $ABCD$.

---

**Annotated GPT-OSS-20B Model Output:** (Reasoning Thought) *"We have rhombus ABCD with vertices on hyperbola x^2/20 - y^2/24 = 1. Diagonals intersect at origin. So origin is intersection of diagonals, meaning the diagonals are perpendicular? In a rhombus, diagonals are perpendicular only if it's a square? Wait rhombus: diagonals perpendicular if it's a kite? Actually in rhombus diagonals are perpendicular only if rhombus is a square? Let's recall: In a rhombus, diagonals are perpendicular if rhombus is a kite? So a rhombus is a kite with equal sides. For a rhombus, diagonals are perpendicular if it's a square? No, square diagonals are perpendicular? In a square, diagonals are perpendicular? Actually square diagonals are perpendicular? Yes, square diagonals are perpendicular? Are square diagonals are perpendicular? In a square, diagonals are perpendicular? Yes they are perpendicular? Let's check:*

*In a square, diagonals are perpendicular? Actually in a square, diagonals intersect at right angles? I think yes they are perpendicular? Wait think: In a square, the diagonals are perpendicular? For a square, the diagonals are perpendicular? I recall that in a square, diagonals are perpendicular? Let's compute: square of side s, diagonals length $s\sqrt{2}$. The angle between diagonals is 90 degrees? In a square, the diagonals are perpendicular? Let's draw: coordinates (0,0),(s,0),(s,s),(0,s). Diagonals: (0,0)-(s,s) line slope 1; other diagonal (s,0)-(0,s) slope -1. They are perpendicular. So yes.\{}n\{}nBut rhombus general: diagonals are perpendicular only if rhombus is a square? Actually rhombus with all sides equal but not necessarily right angles. In general rhombus, diagonals are perpendicular if rhombus is a kite? There is also a rhombus where diagonals are perpendicular but not equal sides?*

(Transition Thought) *Wait property: In a rhombus, diagonals are perpendicular iff rhombus is a square? Let's confirm: In a rhombus, diagonals are perpendicular if rhombus is a kite? Actually a kite has perpendicular diagonals. But a rhombus is a special kite with equal sides. For a rhombus, diagonals are perpendicular only if rhombus is a square? Let's test: Consider rhombus with sides length 1, angles 60 and 120: diagonals lengths: d1 = 2*1*cos(30)=? Wait formula: In rhombus with side a, angle \{}u03b8 between adjacent sides. Diagonals lengths: d1 = a$\sqrt{(2+2cos\{}u03b8)}$ = 2a cos(\{}u03b8/2). d2 = a$\sqrt{(2-2cos\{}u03b8)}$ = 2a sin(\{}u03b8/2). They are perpendicular if product of slopes? Actually if d1*d2? For perpendicular, product of slopes? Wait slope of d1 relative to d2? In rhombus,*

(Execution Thought) *diagonals are perpendicular if d1^2 + d2^2 = 4a^2? Wait maybe property: In rhombus, diagonals are perpendicular iff rhombus is a square? Let's check: For rhombus with \{}u03b8=60°, d1=2a cos(30)=2a*($\sqrt{3}$/2)=a$\sqrt{3}$. d2=2a sin(30)=a. Are they perpendicular? In a square, \{}u03b8=90°, d1=d2=a$\sqrt{2}$. Are they perpendicular? Let's compute slopes: For \{}u03b8=60°, coordinates: place A at (0,0), B at (a,0). Next side AB to BC with angle 60: C = B + (a cos60, a sin60) = (a + a/2, a*$\sqrt{3}$/2). D = A + (a cos60, a sin60) = (a/2, a*$\sqrt{3}$/2). Diagonals: AC from (0,0) to (a + a/2, a*$\sqrt{3}$/2) slope = (a*$\sqrt{3}$/2)/(a+ a/2)= ($\sqrt{3}$/2)/(3/2)=$\sqrt{3}$/3. BD from (a,0) to (a/2, a*$\sqrt{3}$/2) slope = (a*$\sqrt{3}$/2 - 0)/(a/2 - a) = ($\sqrt{3}$/2)/(-a/2)= -$\sqrt{3}$. Product slopes = -3? Not -1. So not perpendicular. So*

*rhombus with \{}u03b8=60 not perpendicular. So only square? Let's test rhombus with \{}u03b8=90: square. Diagonals slopes 1 and -1, product -1. So yes.\{}n\{}nThus rhombus with diagonals perpendicular is a square. But we don't require perpendicular. We just need intersection at origin. So origin is intersection of diagonals; rhombus centered at origin. So vertices*

*are opposite pairs symmetrical about origin. So if A at (x1,y1), then C at (-x1,-y1). Similarly B at (x2,y2), D at (-x2,-y2). All four points lie on hyperbola.\{}n\{}nThus we have points (x1,y1), (x2,y2) such that both satisfy hyperbola equation: x^2/20 - y^2/24 = 1.\{}n\{}nWe need to find maximum possible BD^2?*

*Wait we need largest number less than BD^2 for all rhombus? Wait question: \{}"find the largest number less than BD^2 for all rhombus ABCD\{}". Means we need minimal possible BD^2 Wait \{}"largest number less than BD^2 for all rhombus\{}" ambiguous. Let's parse: We want the largest number less than BD^2 for all rhombus? Means we consider all possible rhombuses, compute BD^2, then find the maximum possible BD^2? But they ask \{}"largest number less than BD^2 for all rhombus\{}" maybe they want supremum of BD^2? Actually \{}"largest number less than BD^2 for all rhombus\{}" could mean find maximum possible BD^2? Wait \{}"largest number less than BD^2 for all rhombus\{}" could mean find the greatest lower bound? Let's read: \{}"find the largest number less than BD^2 for all rhombus ABCD\{}". Might mean find the maximum possible value of BD^2? But phrase \{}"less than BD^2\{}" maybe they want the largest value that is less than any*

*BD^2? That would be infimum? Wait \{}"largest number less than BD^2 for all rhombus\{}" maybe they want the maximum of BD^2? Let's analyze.\{}n\{}nWe need to interpret: \{}"largest number less than BD^2 for all rhombus ABCD\{}". Suppose we want to find a number L such that for all rhombus, BD^2 ¿ L. And L is as large as possible. That is essentially the infimum of BD^2 over all rhombus? Wait we want largest lower bound? Actually \{}"less than BD^2\{}" meaning L ¡ BD^2 for all rhombus. So L is a lower bound of the set of BD^2 values. Largest such L is the infimum (greatest lower bound). So they ask for infimum of BD^2? Wait \{}"largest number less than BD^2\{}" might mean the greatest number that is less than*

*BD^2 for all rhombus? That is indeed infimum. But they might ask for maximum possible BD^2? Let's check: Suppose BD^2 can vary. They want largest number less than BD^2? If BD^2 can be arbitrarily large, then largest number less than BD^2 is infinite. But maybe BD^2 has a maximum? Let's compute BD^2 formula maybe depends on angles; maybe BD^2 has maximum. But they ask \{}"largest number less than BD^2 for all rhombus\{}". That seems weird. Let's compute BD^2 as a function of something. Perhaps BD^2 can be arbitrarily large? Let's check: Points on hyperbola; as x large, y small. But y must satisfy hyperbola. For given x, y^2 = 24*(x^2/20 -1). So y ~ sqrt(24*(x^2/20 -1)). For large —x—, y grows roughly sqrt(24/20) —x— = sqrt(6/5)—x— ~ 1.095—x—.*

*So as —x— large, y ~ 1.095—x—. So coordinates far away produce large distances. BD^2 between points B and D? B at (x2,y2), D at (-x2,-y2). So BD vector = (-x2 - x2, -y2 - y2) = (-2x2, -2y2). So BD^2 = 4(x2^2 + y2^2). That's 4*(x2^2 + y2^2). Since x2,y2 satisfy hyperbola, we can compute x2^2 + y2^2 variable. As —x2— large, y2 also large, so BD^2 large. So no maximum; so maybe they want minimal BD^2? Let's read again: \{}"find the largest number less than BD^2 for all rhombus ABCD.\{}" Maybe they mean find the largest number that is less than every BD^2. That is the greatest lower bound? That would be minimal BD^2?*

*The largest number less than all BD^2 would be the maximum lower bound = infimum. So they ask for infimum of BD^2 across all rhombus. That is the minimal possible BD^2. So we need to find minimal BD^2. So they want largest number less than BD^2 for all rhombus, i.e., the greatest lower bound. Equivalent to minimal BD^2? If BD^2 has minimum, then that minimum is greater than all smaller numbers; but largest number less than BD^2 for all rhombus would be the minimum BD^2 itself? Actually if BD^2 has a minimum value m, then for all rhombus, BD^2 ¿= m. Then m is less than or equal to all BD^2. But largest number less than BD^2 for all rhombus would be m? But m is not less than BD^2 for rhombus that achieve m; it's equal. But phrase*

*\{}"less than BD^2\{}" maybe they mean \{}"less than or equal to\{}"? ambiguous.\{}n\{}nLet's analyze: If we want minimal BD^2, we can compute BD^2 = 4(x^2 + y^2). For point (x,y) on hyperbola. So BD^2 = 4(x^2 + y^2). We need to minimize this subject to hyperbola equation. So we need to find minimal value of x^2 + y^2 given x^2/20 - y^2/24 = 1. That is a constrained optimization.\{}n\{}nSet x^2 = 20(1 + y^2/24). Hyperbola: x^2/20 - y^2/24 = 1 =¿ x^2 = 20 + (20/24) y^2 = 20 + (5/6) y^2. So x^2 + y^2 = 20 + (5/6)y^2 + y^2 = 20 + (11/6)y^2. So BD^2 = 4(20 + (11/6)y^2) = 80 + (44/6) y^2 = 80 + (22/3) y^2.\{}n\{}nThus BD^2 depends on y^2. Since y^2*

¿ = 0, BD^2 ¿ = 80. So minimal BD^2 is 80 when y=0. But point with y=0 lies on hyperbola? Plug y=0: x^2/20 =1 =¿ x^2=20 =¿ x=\{}u00b1√20. So vertices at (\{}u00b1√20,0). Then A and C at (√20,0) and (-√20,0). B and D at (x2,y2) and (-x2,-y2). But if y=0, then B and D at (x2,0) and (-x2,0). But then all four points are collinear? If y=0 for B and D, then B and D on same horizontal line as A and C? Actually A at (√20,0), C at (-√20,0). So all four vertices would be on same line, degenerating rhombus? That would not be a rhombus. But we must have non-zero y2 to have rhombus shape? Let's think: For a rhombus, we need four distinct points not collinear.

If B and D share same y=0, then B and D are on same horizontal line as A and C; but then the rhombus would have zero area? Actually if all points lie on same line, then shape degenerates. So we need y2 ≠ 0. So minimal BD^2 might be ¿ 80.\{}n\{}nWe need to ensure that A,B,C,D form a rhombus: all sides equal. But we only used that diagonals intersect at origin; we need to ensure rhombus condition: AB=BC=CD=DA. We used only that A and C are opposite, B and D opposite. But need side lengths equal. So we need to impose side equality.\{}n\{}nLet coordinates: A=(x1,y1), C=(-x1,-y1). B=(x2,y2), D=(-x2,-y2). Then side AB: vector B-A = (x2 - x1, y2 - y1). Length squared L1 = (x2 -

x1)^2 + (y2 - y1)^2.\{}n\{}nBC: C-B = (-x1 - x2, -y1 - y2). Length squared L2 = (-x1 - x2)^2 + (-y1 - y2)^2 = (x1 + x2)^2 + (y1 + y2)^2.\{}n\{}nCD: D-C = (-x2 + x1, -y2 + y1) = (x1 - x2, y1 - y2). Length squared L3 = (x1 - x2)^2 + (y1 - y2)^2 = same as L1? L3 equals L1? Actually AB and CD are opposite sides; they should be equal. Similarly BC and DA equal.\{}n\{}nBut for rhombus, all four side lengths equal: L1 = L2 = L3 = L4. But due to symmetry, L1 = L3 automatically? Let's check: L1 = (x2 - x1)^2 + (y2 - y1)^2. L3 = (x1 - x2)^2 + (y1 - y2)^2 = same as L1.

So L1 = L3. Similarly, L2 = (x1 + x2)^2 + (y1 + y2)^2. L4 = (x2 + x1)^2 + (y2 + y1)^2 = same as L2. So we need L1 = L2.\{}n\{}nThus condition: (x2 - x1)^2 + (y2 - y1)^2 = (x1 + x2)^2 + (y1 + y2)^2).\{}n\{}nExpand: LHS = (x2 - x1)^2 + (y2 - y1)^2 = x2^2 - 2x1x2 + x1^2 + y2^2 - 2y1y2 + y1^2.\{}n\{}nRHS = (x1 + x2)^2 + (y1 + y2)^2 = x1^2 + 2x1x2 + x2^2 + y1^2 + 2y1y2 + y2^2.\{}n\{}nSet equal: LHS = RHS.\{}n\{}nSimplify: LHS minus RHS = (x2^2 - 2x1x2 + x1^2 + y2^2 - 2y1y2 + y1^2) - (x1^2 + 2x1x2 + x2^2 + y1^2 + 2y1y2 + y2^2) = 0.\{}n\{}nCancel x2^2, x1^2, y2^2, y1^2. Left

with (-2x1x2 - 2y1y2) - (2x1x2 + 2y1y2)?? Compute:\{}n\{}n-2x1x2 - 2y1y2 - (2x1x2 + 2y1y2) = -2x1x2 - 2y1y2 - 2x1x2 - 2y1y2 = -4x1x2 - 4y1y2.\{}n\{}nSet equal to 0: -4x1x2 - 4y1y2 = 0 =¿ x1x2 + y1y2 = 0.\{}n\{}nThus rhombus condition: x1x2 + y1y2 = 0.\{}n\{}nBut we also have that all points on hyperbola: x1^2/20 - y1^2/24 = 1; x2^2/20 - y2^2/24 = 1.\{}n\{}nWe need to find solutions (x1,y1),(x2,y2) satisfying x1x2 + y1y2 = 0.\{}n\{}nWe also want BD^2 = 4(x2^2 + y2^2). We need to minimize BD^2 subject to constraints.\{}n\{}nLet's denote variables: Let coordinates for A: (x1,y1). For B: (x2,y2). Condition: x1x2 + y1y2 = 0.\{}n\{}nWe can param using angles? Since hyperbola: paramization: x = √20 cosh t, y = √24 sinh t? Hyperbola

standard: x^2/20 - y^2/24 = 1. Use param: x = √20 cosh u, y = √24 sinh u. Because cosh^2 - sinh^2 =1. So indeed x^2/20 = cosh^2 u, y^2/24 = sinh^2 u. So coordinates: (√20 cosh u, √24 sinh u). Similarly for second point: (√20 cosh v, √24 sinh v). But we also need x2^2/20 - y2^2/24 =1. So both points satisfy.\{}n\{}nThus x1 = √20 cosh u, y1 = √24 sinh u. x2 = √20 cosh v, y2 = √24 sinh v.\{}n\{}nNow rhombus condition: x1x2 + y1y2 = 0.\{}n\{}nCompute: x1x2 + y1y2 = √20 cosh u * √20 cosh v + √24 sinh u * √24 sinh v = 20 cosh u cosh v + 24 sinh u sinh v.\{}n\{}nSet equal to 0: 20 cosh u cosh

v + 24 sinh u sinh v = 0.\{}n\{}nDivide by 4: 5 cosh u cosh v + 6 sinh u sinh v = 0.\{}n\{}nWe can use identity: cosh(u+v) = cosh u cosh v + sinh u sinh v. But coefficients differ. Maybe use hyperbolic identity: cosh(u+v) = cosh u cosh v + sinh u sinh v. Similarly for v. Then compute:\{}n\{}ncosh u cosh v = 1/√((1-a^2)(1-b^2)). sinh u sinh v = ab / √((1-a^2)(1-b^2). So equation:\{}n\{}n5 * 1/√((1-a^2)(1-b^2))

+ 6 * ab / √((1-a^2)(1-b^2)) = 0.\{}n\{}nMultiply both sides by √((1-a^2)(1-b^2)):\{}n\{}n5 + 6ab = 0 =¿ ab = -5/6.\{}n\{}nThus tanh u * tanh v = -5/6.\{}n\{}nThus we have relation between u and v. Since tanh u in (-1,1). So product negative implies u and v of opposite signs. So we can pick u¿0, v¡0 etc.\{}n\{}nThus we have relation: tanh u * tanh v = -5/6.\{}n\{}nNow we need BD^2 = 4(x2^2 + y2^2) = 4(20 cosh^2 v + 24 sinh^2 v) = 4(20 cosh^2 v + 24 sinh^2 v). But cosh^2 - sinh^2

=1. So cosh^2 = 1 + sinh^2. So 20 cosh^2 v + 24 sinh^2 v = 20(1 + sinh^2 v) + 24 sinh^2 v = 20 + (20+24) sinh^2 v = 20 + 44 sinh^2

v.\{}n\{}nThus BD^2 = 4(20 + 44 sinh^2 v) = 80 + 176 sinh^2 v.\{}n\{}nSimilarly we could express in terms of tanh v. Since sinh^2 v = tanh^2 v * cosh^2 v. But maybe easier: Use identity: sinh^2 v = (cosh^2 v -1). But we can express cosh^2 v in terms of tanh^2: cosh^2 = 1/(1 - tanh^2). So sinh^2 = cosh^2 -1 = (1/(1 - t^2) -1) = (1 - (1 - t^2))/ (1 - t^2) = t^2/(1 - t^2). Where t = tanh v.\{}n\{}nThus BD^2 = 4(20 + 44 * t^2/(1 - t^2)). So BD^2 = 80 + 176 * t^2/(1 - t^2).\{}n\{}nNow we need to minimize BD^2 over possible t satisfying relation with u: tanh u * tanh v = -5/6. But t = tanh v.

Let a = tanh u. Then a*b = -5/6 where b = tanh v = t. So t = -5/(6a). Since a in (-1,1). So t must satisfy —t— ¡1. So need — -5/(6a)— ¡ 1 =¿ —5/(6a)— ¡ 1 =¿ —a— ¿ 5/6. So a must be between 5/6 and 1 or -1 and -5/6. But a is tanh u; so u large magnitude ¿? tanh ¿5/6 implies —u— ¿ artanh(5/6). So a in (5/6,1) or (-1,-5/6). But since a and t have opposite signs? Because product negative, if a¿0 then t¡0, etc. So we can consider a¿5/6, t negative.\{}n\{}nNow we need to compute BD^2 in terms of a? Because t expressed in a. But BD^2 depends only on t (since BD^2 formula). So we need

to minimize BD^2 over t satisfying relation? Wait BD^2 depends only on t. But t = tanh v. But v's relation to u ensures t = -5/(6a). So we can vary a in (5/6,1). For each a, we have t = -5/(6a). Then BD^2 = 80 + 176 * t^2/(1 - t^2). Let's compute expression in a.\{}n\{}nLet t = -5/(6a). Then t^2 = 25/(36 a^2). Compute 1 - t^2 = 1 - 25/(36 a^2) = (36 a^2 - 25)/(36 a^2). Then t^2/(1 - t^2) = (25/(36 a^2)) / ((36 a^2 -25)/(36 a^2)) = 25/(36 a^2) * (36 a^2)/(36 a^2 -25) = 25/(36 a^2 -25).\{}n\{}nThus BD^2 = 80 + 176 * 25/(36 a^2 - 25) = 80 + (176*25)/(36 a^2 - 25). Simplify: 176*25 = 4400. So BD^2 =

80 + 4400/(36 a^2 - 25).\{}n\{}nNow a in (5/6,1). We need to minimize BD^2? Actually BD^2 increases as denominator decreases. Denominator 36 a^2 -25. Since a^2 ¡1, denominator ranges? For a close to 1, denominator = 36*1 -25 = 11. For a close to 5/6, a^2 = 25/36, denominator = 36*(25/36)-25 = 25-25=0. So denominator goes to 0 from positive side? So a^2 = 25/36 =¿ 36 a^2 =25 =¿ denominator = 0. So as a approaches 5/6, denominator tends to 0+, so BD^2 tends to infinity. So BD^2 minimal occurs at largest denominator, i.e., a=1. So minimal BD^2 occurs when a=1 (tanh u =1). But tanh u cannot be exactly 1? As u\{}u2192\{}u221e, tanh u\{}u21921. So a can approach 1 from below. Then denominator=36*1 -25=11. So BD^2

= 80 + 4400/11 = 80 + 400 = 480. 4400/11 = 400. So BD^2 tends to 480 as a\{}u21921. So minimal BD^2 = 480? But we need to confirm that a=1 corresponds to u infinite, but we still need to satisfy rhombus condition with finite coordinates? As u\{}u2192\{}u221e, point A tends to ($\sqrt{20}$ cosh u, $\sqrt{24}$ sinh u). As u\{}u2192\{}u221e, cosh u ˜ (e^u)/2 large. So coordinates huge. But we also have v such that tanh v = -5/(6a) ˜ -5/6. So v finite negative such that tanh v = -5/6. So B coordinates finite. So rhombus with one vertex at infinity? A at infinity, B finite. But rhombus with infinite vertex? Not allowed. So we cannot let a\{}u21921 exactly? Actually a=1 means u\{}u2192\{}u221e, so

A's coordinates infinite. So not allowed. But we can approach arbitrarily close to 1, giving arbitrarily large coordinates for A. But we need finite coordinates for all vertices. So minimal BD^2 may not be achievable but we can approach 480 from above. So largest number less than BD^2 for all rhombus is 480? Wait we need largest number less than BD^2 for all rhombus. Since BD^2 ¿ 480 for all finite rhombus? Let's test: For any finite u, a ¡1, BD^2 ¿ 480? Since BD^2 = 80 + 4400/(36 a^2 - 25). Denominator ¿ 0. As a increases, denominator increases, BD^2 decreases. So BD^2 minimal when denominator maximum. Denominator increases with a^2. So maximum a^2 ¡1. So minimal BD^2 ¿ 80 + 4400/(11)?? Wait compute at a close

to 1: a=0.999? Let's compute approximate: a^2=0.998001, denom = 36*0.998001 -25 = 35.928036 -25=10.928036. BD^2 = 80 + 4400/10.928036 = 80 + 402.9 = 482.9. So BD^2 ¿ 480. So minimal BD^2 ¿ 480. So largest number less than BD^2 for all rhombus is 480? But BD^2 always ¿ 480? Actually at a close to 1, BD^2 ˜482.9 ¿ 480. So BD^2 ¿ 480 for all finite rhombus. So the greatest lower bound is 480? But BD^2 ¿ 480 for all rhombus. So 480 is less than BD^2 for all rhombus. But is there any rhombus with BD^2 arbitrarily close to 480? As a\{}u21921, BD^2 tends to 480 from above. So 480 is the greatest lower bound. So answer: 480.\{}n\{}nBut they ask \{}"largest number less than

*BD^2 for all rhombus\{}" meaning the supremum of numbers less than all BD^2. That is 480. So answer: 480.\{}n\{}nLet's double-check with example: choose u large but finite, say u=5, tanh u ~ 0.9999? Actually tanh 5 ~ 0.999909. a^2 ~ 0.9998. Denominator = 36*0.9999 -25 = 35.996 -25=10.996. BD^2 = 80 + 4400/10.996 = 80 + 400.6 = 480.6. So BD^2 ~ 480.6 ¿ 480. So indeed.\{}n\{}nThus answer: 480.\{}n\{}nWe should confirm rhombus condition: x1x2 + y1y2 = 0. Let's test with u large, v finite such that tanh v = -5/(6 tanh u). For u=5, a=0.9999, t= -5/(6*0.9999) ≈ -0.8334. Then v = artanh(-0.8334) ≈ -1.0? Actually tanh(-1) = -0.761. Not -0.833. So need v such that tanh v = -0.833. v ≈ -1.2? Let's compute tanh(-1.2) ≈*

*-0.833. Yes. So coordinates: A at (x1,y1) huge, B at finite. So rhombus vertices: A large, B finite, C opposite of A large negative, D opposite of B. So rhombus large, but still finite? Actually A coordinates huge, but still finite. So rhombus defined. So BD^2 = 4*(x2^2 + y2^2). x2 finite, y2 finite. So BD^2 finite. So rhombus exists. So BD^2 ¿ 480. So answer 480.\{}n\{}nThus final answer: \{}\{}boxed{480}.assistantfinalLet the vertices of the rhombus be \{}n\{}n\{}\{}[\{}nA=(x_1,y_1),\{}\{}quad B=(x_2,y_2),\{}\{}quad C=(-x_1,-y_1),\{}\{}quad D=(-x_2,-y_2)\{}n\{}\{}]\{}n\{}nAll four points lie on the hyperbola\{}n\{}n\{}\{}[\{}n\{}\{}frac{x^{2}}{20}-\{}\{}frac{y^{2}}{24}=1 .\{}n\{}\{}]\{}n\{}n\{}n### 1. Parameterisation of the hyperbola\{}n\{}nWrite\{}n\{}n\{}\{}[\{}nx=\{}\{}sqrt{20}\{}\{}cosh t,\{}\{}qquad y=\{}\{}sqrt{24}\{}\{}sinh t\{}n\{}\{}]\{}n\{}nfor a real parameter \{}\{}(t\{}\{}). \{}nThus\{}n\{}n\{}\{}[\{}nA=(\{}\{}sqrt{20}\{}\{}cosh u,\{}\{};\{}\{}sqrt{24}\{}\{}sinh u),\{}\{}qquad \{}nB=(\{}\{}sqrt{20}\{}\{}cosh v,\{}\{};\{}\{}sqrt{24}\{}\{}sinh v).\{}n\{}\{}]\{}n\{}n\{}n### 2. Rhombus condition\{}n\{}nThe sides of a rhombus are equal, so\{}n\{}n\{}\{}[\{}n—(x_2-x_1)^2+(y_2-y_1)^2—=(x_1+x_2)^2+(y_1+y_2)^2 .\{}n\{}\{}]\{}n\{}nThis simplifies to\{}n\{}n\{}\{}[\{}nx_1x_2+y_1y_2=0. \{}\{}tag{1}\{}n\{}\{}]\{}n\{}nSubstituting the hyperbolic parametrisation gives\{}n\{}n\{}\{}[\{}n\{}\{}sqrt{20}\{}\{}cosh u\{}\{};\{}\{}sqrt{20}\{}\{}cosh v+\{}\{}sqrt{24}\{}\{}sinh u\{}\{};\{}\{}sqrt{24}\{}\{}sinh v=0\{}n\{}\{}]\{}n\{}nor\{}n\{}n\{}\{}[\{}n5\{}\{}cosh u\{}\{}cosh v+6\{}\{}sinh u\{}\{}sinh v=0. \{}\{}tag{2}\{}n\{}\{}]\{}n\{}nUsing \{}\{}(\{}\{}tanh u=\{}\{}dfrac{\{}\{}sinh u}{\{}\{}cosh u}\{}\{}) and \{}\{}(\{}\{}tanh v=\{}\{}dfrac{\{}\{}sinh v}{\{}\{}cosh v}\{}\{}),\{}nequation (2) becomes\{}n\{}n\{}\{}[\{}n5\{}\{}tanh u\{}\{},\{}\{}tanh v+6=0 \{}n\{}\{}quad\{}\{}Longrightarrow\{}\{}quad\{}n\{}\{}tanh u\{}\{},\{}\{}tanh v=-\{}\{}frac56 . \{}n\{}\{}tag{3}\{}n\{}\{}]\{}n\{}n### 3. Diagonal length \{}\{}(BD\{}\{})\{}n\{}nThe diagonal \{}\{}(BD\{}\{}) joins \{}\{}(B\{}\{}) and \{}\{}(D\{}\{}), so\{}n\{}n\{}\{}[\{}nBD^2=4(x_2^2+y_2^2)\{}n =4\{}\{}bigl(20\{}\{}cosh^2 v+24\{}\{}sinh^2 v\{}\{}bigr)\{}n =4\{}\{}Bigl(20+44\{}\{}sinh^2 v\{}\{}Bigr).\{}n\{}\{}]\{}n\{}nLet \{}\{}(t=\{}\{}tanh v\{}\{}). \{}nThen \{}\{}(\{}\{}sinh^2 v=\{}\{}dfrac{t^2}{1-t^2}\{}\{}), and\{}n\{}n\{}\{}[\{}nBD^2=4\{}\{}Bigl(20+44\{}\{}frac{t^2}{1-t^2}\{}\{}Bigr)\{}n =80+\{}\{}frac{176\{}\{},t^2}{1-t^2}. \{}\{}tag{3a}\{}n\{}\{}]\{}n\{}n### 4. Express \{}\{}(t\{}\{}) through \{}\{}(\{}\{}tanh u\{}\{})\{}n\{}nFrom (3) we have \{}\{}(t=-\{}\{}dfrac{5}{6\{}\{},\{}\{}tanh u}\{}\{}). \{}nLet \{}\{}(a=\{}\{}tanh u\{}\{};(0¡a¡1)\{}\{}). Then\{}n\{}n\{}\{}[\{}nt^2=\{}\{}frac{25}{36a^2} \{}\{}quad\{}\{}Longrightarrow\{}\{}quad\{}n\{}\{}frac{t^2}{1-t^2}=\{}\{}frac{25}{36a^2-25}.\{}n\{}\{}]\{}n\{}nSubstituting into (3a),\{}n\{}n\{}\{}[\{}nBD^2=80+\{}\{}frac{4400}{36a^2-25}. \{}\{}tag{4}\{}n\{}\{}]\{}n\{}n### 5. Minimising \{}\{}(BD^2\{}\{})\{}n\{}nFor a rhombus we need \{}\{}(0¡a¡1\{}\{}). \{}nThe denominator \{}\{}(36a^2-25\{}\{}) increases with \{}\{}(a\{}\{}), so \{}\{}(BD^2\{}\{}) is\{}n**minimal** when \{}\{}(a\{}\{}) is as large as possible, i.e. as \{}\{}(a\{}\{}to 1^{-}\{}\{}).\{}n\{}nTaking the limit \{}\{}(a\{}\{}to 1\{}\{}),\{}n\{}n\{}\{}[\{}n\{}\{}lim_{a\{}\{}to1}BD^2=80+\{}\{}frac{4400}{11}=480 .\{}n\{}\{}]\{}n\{}nFor every finite rhombus \{}\{}(a¡1\{}\{}),\{}\{};BD^2¿480\{}\{}); hence\{}n\{}\{}(480\{}\{}) is the greatest lower bound of \{}\{}(BD^2\{}\{}).\{}n\{}n\{}n\{}\{}[\{}n\{}\{}\{}boxed{480}]",*

---

**Input Prompt 2:** Every morning Aya goes for a 9-kilometer-long walk and stops at a coffee shop afterwards. When she walks at a constant speed of $s$ kilometers per hour, the walk takes her $4$ hours, including $t$ minutes spent in the coffee shop. When she walks at $s + 2$ kilometers per hour, the walk takes her 2 hours and $24$ minutes, including $t$ minutes spent in the coffee shop. Suppose Aya walks at $s + \frac{1}{2}$ kilometers per hour. Find the number of minutes the walk takes her, including the $t$ minutes spent in the coffee shop.

---

**GPT-OSS-20B Model Output:** *We have a problem: Aya walks 9 km, stops at coffee shop for $t$ minutes. When walking at speed $s$ km/h, time walking $= 9/s$ hours. Total time including coffee shop $= 9/s + t/60$ hours $= 4$ hours. So equation (1): $9/s + t/60 = 4$. When walking at speed $s + 2$ km/h, walking time $= 9/(s + 2)$. Total time $= 9/(s + 2) + t/60 = 2$ hours 24 minutes $= 2 + 24/60 = 2 + 0.4 = 2.4$ hours. So equation (2): $9/(s + 2) + t/60 = 2.4$. We need $t$ and $s$. Solve. Let's denote $t_{min} = t$ (minutes). Convert $t/60$ to hours. Use equations. Equation (1): $9/s = 4 - t/60$. Equation (2): $9/(s + 2) = 2.4 - t/60$. Let's denote $A = t/60$. Then: $9/s = 4 - A \Rightarrow s = 9/(4 - A)$. (Equation A) $9/(s + 2) = 2.4 - A \Rightarrow s + 2 = 9/(2.4 - A) \Rightarrow s = 9/(2.4 - A) - 2$. (Equation B) Set them equal: $9/(4 - A) = 9/(2.4 - A) - 2$. Solve for A. Let's compute. Multiply both sides by denominators. Let's let $x = A$. Equation: $9/(4 - x) = 9/(2.4 - x) - 2$. Bring terms: $9/(4 - x) + 2 = 9/(2.4 - x)$. Compute LHS: $9/(4 - x) + 2 = [9 + 2(4 - x)]/(4 - x) = [9 + 8 - 2x]/(4 - x) = (17 - 2x)/(4 - x)$. Thus RHS: $9/(2.4 - x)$. So $(17 - 2x)/(4 - x) = 9/(2.4 - x)$. Cross-multiply: $(17 - 2x)(2.4 - x) = 9(4 - x)$. Compute left side: Expand. Let's compute $(17 - 2x)(2.4 - x)$. Multiply: $17 \cdot 2.4 = 40.8$. $17 \cdot (-x) = -17x$. $(-2x) \cdot 2.4 = -4.8x$. $(-2x)(-x) = 2x^2$. Sum: $40.8 - 17x - 4.8x + 2x^2 = 40.8 - 21.8x + 2x^2$. Right side: $9(4 - x) = 36 - 9x$. Set equal: $40.8 - 21.8x + 2x^2 = 36 - 9x$. Bring all to left: $40.8 - 21.8x + 2x^2 - 36 + 9x = 0 \Rightarrow (40.8 - 36) = 4.8$. Combine x terms: $-21.8x + 9x = -12.8x$. So equation: $2x^2 - 12.8x + 4.8 = 0$. Divide by 2: $x^2 - 6.4x + 2.4 = 0$. Let's just solve: $x^2 - 6.4$? Let's just solve $9/(4$? I'm stuck. Let's just solve: $x^2 - 6.4$? I'm stuck. Let's just solve the equation: $x^2 - 6.4$. Eh. Ok, let's solve: $x^2 - 6.4$? I'm stuck. Let's just compute: $x^2 - 6.4$? I'm messing. Ok, just do step by direct solving. Equation: $x^2 - 6.4$? I'm messing. Let's just solve the quadratic: $x^2 - 6.4$? Let's solve: $x^2 - 6$? I'm stuck. . . . I'm failing. Ok I'm messing. I'm stuck. I'm failing. I realize I'm messing. I'm stuck. I'm failing. Ok, I'm failing. I'm stuck. I'm stuck. Ok, just compute: $x^2 - 6$? I'm failing. I'm stuck. I'm stuck. This is failing. Ok, I'm going to solve: $x^2 - 6$? I'm stuck. This is failing. I'm stuck. I'm stuck. I'm failing.*

**Note:** The example above illustrates that if a transition thought is fully evicted or quantized to 0 bits, the model enters an endless loop, underscoring that the notional presence of transition thoughts is critical for stable generation.

## F    LIMITATIONS

While ThinKV demonstrates strong performance for long-output reasoning tasks, it is not directly applicable to settings dominated by long input contexts. Should future LRMs place greater emphasis on long-input contexts, additional exploration will be required.

## G    IMPACT STATEMENT

This work improves the generation efficiency of large reasoning models (LRMs) by compressing the KV cache, substantially reducing memory overhead while preserving reasoning accuracy. This enables continuous long-output generation without out of memory (OOM) failures and supports larger batch sizes, yielding higher throughput. Beyond reducing memory, our method maximizes efficiency, contributing to more sustainable AI deployment and expanding accessibility to commodity hardware. As LRMs scale to produce longer outputs, KV cache compression remains an underexplored yet critical direction; our framework offers a generalizable solution that may inspire future algorithm–system co-design. Importantly, while enhancing efficiency, our method introduces no additional societal risks beyond those inherent to LRMs.

## H    LLM USAGE STATEMENT

Portions of this paper were refined with the assistance of a large language model (LLM), specifically ChatGPT 5, used exclusively to polish writing and help reduce verbosity to meet page limit. All technical content, methods, and results were conceived and developed entirely by the authors, without influence from any AI tool.

