# OpenReview forum: "ThinKV: Thought-Adaptive KV Cache Compression for Efficient Reasoning Models"
_ICLR.cc/2026/Conference — ICLR 2026 Oral_

### Official Review · Reviewer_86j3 · 2025-10-21

**Soundness:** 3
**Presentation:** 3
**Contribution:** 3
**Rating:** 6
**Confidence:** 4

**Summary:**

This paper introduces a novel approach to KV Cache management specifically designed for Large Reasoning Models (LRMs). Recognizing that these models generate extremely long sequences, leading to significant KV Cache demands, the authors address that existing methods often overlook the unique characteristics of reasoning models. The core contribution lies on observations about attention sparsity variations across different reasoning stages. Based on these findings, the proposed method, ThinKV, dynamically manages the KV Cache through a combination of quantization and eviction strategies. This approach aims to simultaneously reduce latency and improve peak throughput, enhancing the overall efficiency of LRM inference.

**Strengths:**

* **A Novel Design.** The paper introduces a thought-adaptive framework that maps attention sparsity to reasoning stages, enabling hybrid mixed-precision quantization and adaptive eviction. This design achieves fine-grained, semantically informed KV cache compression beyond traditional token-level compressions.
* **A Comprehensive Implementation.** ThinKV is supported by a strong system-level realization, featuring gather-free memory reuse via an extension from vLLM mechanism and fused dequantization–matrix multiplication inspired by KiVi. The paper further provides detailed kernel-level breakdowns and overhead analyses, demonstrating solid engineering rigor.
* **A Strong Result.** Across math and coding benchmarks, ThinKV consistently achieves higher accuracy under different memory budgets and higher throughput compared to R-KV, showcasing both algorithmic and system-level efficiency gains.

**Weaknesses:**

* **Lack of Latency Comparison.** Although Section 5 reports throughput under large-batch settings, the paper does not present a detailed latency analysis. For large reasoning models (LRMs), metrics especially TPOT (Time per Output Token) are critical, as generated sequence will be long and interactive responsiveness matters. A fine-grained latency breakdown across decoding steps or batch sizes would strengthen the system-level evaluation.
* **Limited on long input tasks.** ThinKV is evaluated on datasets whose input prompts average only a few hundred tokens. However, some practical workloads (e.g., SWE-bench [1]) involve very long input contexts. Since ThinKV does not compress the prefill cache, its effectiveness and efficiency under long-input regimes remain unclear.



[1] Jimenez, Carlos E., et al. "SWE-bench: Can Language Models Resolve Real-world Github Issues?." ICLR. 2024.

**Questions:**

* **Accuracy–Efficiency Trade-off with Non-Reasoning Modes.**
   While ThinKV improves accuracy relative to other compression baselines, it still introduces slight degradation compared to Full Cache. Recent models such as the Qwen3 series support *optional reasoning (thinking vs. direct answer)* modes. It would be valuable to compare ThinKV-enabled reasoning against no-thinking (full attention) decoding, to quantify whether the added reasoning—and compression—yields net gains in both accuracy and efficiency.
* **Sensitivity of different Thought Types to Quantization Precision.**
   Section 3.2.2 discusses precision settings on different reasoning stages. Prior work such as QAQ [2] demonstrates that Key and Value Cache have different sensitivity on precision. A similar quantization-sensitivity study across thought types would clarify the boundary of compression on different reasoning stages.

[2] Dong, Shichen, et al. "Qaq: Quality adaptive quantization for llm kv cache." arXiv preprint arXiv:2403.04643 (2024).

---

> ### Author Response · Authors · 2025-11-20
> **Author Response #1 to Reviewer 86j3**
>
> We sincerely thank the reviewer for taking the time to provide a thoughtful and constructive evaluation of our work. We greatly appreciate the recognition of our thought-adaptive design (TBQ and TBE), system-level implementation rigor, and strong empirical results across math and code reasoning tasks. We address the reviewer’s questions and concerns in detail below. For clarity, we mark responses to weakness comments with **W#** and responses to explicit questions with **Q#**.
>
> ---
>
> ## **Q1: Compare ThinKV-enabled reasoning with no-thinking (full-attention) decoding on Qwen3 model family**
>
> Thank you raising this important point. Following your suggestions, we employ Qwen3-8B[1] as the representative model for this study. This model conveniently supports both *thinking* and *non-thinking* modes through the `/think` and `/no_think` flags. We summarize our results in Table A below and have also updated the manuscript with these results in Appendix E.13.
>
> **Table A:** Accuracy comparison between thinking, non-thinking modes and ThinKV enabled thinking mode on Qwen3-8B evaluated on LiveCodeBench.
>
> | **Method**              | **Mode**         | **Avg. Precision / Eviction Budget**   | **Accuracy (\%)** |
> |-------------------------|------------------|----------------------------------------|-------------------|
> | FullKV                  | Non-Thinking     | –                                      | 21.8              |
> | FullKV                  | Thinking         | –                                      | 55.6              |
> | ThinKV                  | Thinking         | 3.6 / 1024                             | 53.4              |
> | ThinKV                  | Thinking         | 3.7 / 2048                             | 55.2              |
>
>
> Evidently, ThinKV in thinking mode incurs ≤2.2\% accuracy drop across different eviction budgets while using <6.87\% of the FullKV memory. In contrast, the non-thinking mode suffers a severe (>33\%) accuracy degradation, highlighting that reasoning-augmented decoding is necessary for correctness.
>
> Along the same lines, GPT-OSS also exposes a reasoning-effort knob (with low, medium, and high settings), and in Appendix E.7 of the revised manuscript we provide additional results evaluating ThinKV across these different reasoning-effort levels.
>
> [1] Yang, An, et al. "Qwen3 technical report." arXiv preprint arXiv:2505.09388 (2025).
>
> ---

---

> ### Author Response · Authors · 2025-11-20
> **Author Response #2 to Reviewer 86j3**
>
> ## **Q2: QAQ-like quantization-sensitivity analysis across various thought types**
>
> Thank you for sharing the reference to the QAQ paper. Following your suggestion, we have conducted a QAQ-style  KV quantization-sensitivity analysis across the three thought types identified in our work—Reasoning, Execution, and Transition. Specifically, for simplicity, we conduct this analysis using uniform INT quantization. When sweeping the precision of K or V within a given thought type, all remaining KV-cache elements of all thought types are held fixed at 8-bit, ensuring that the observed accuracy changes stem solely from the targeted quantization setting. In Tables B–D below, we independently sweep the K and V precisions from 1 to 8 bits for each thought type and report the corresponding accuracy trends for R1-Llama-70B on LiveCodeBench. We also present the above results as a plot in Appendix E.9 of the revised manuscript.
>
> **Table B:** Quantization Sensitivity for Reasoning Thoughts
> | **K Precision** | **Accuracy (\%)** | **V Precision** | **Accuracy (\%)** |
> |----------------|------------------|------------------|------------------|
> | 1              | 0                | 1                | 0                |
> | 2              | 0                | 2                | 11.5             |
> | 3              | 27.3             | 3                | 38.5             |
> | 4              | 51.6             | 4                | 52.94            |
> | 5              | 52.94            | 5                | 52.94            |
> | 6              | 52.94            | 6                | 52.94            |
> | 7              | 52.94            | 7                | 52.94            |
> | 8              | 52.94            | 8                | 52.94            |
>
> **Table C:** Quantization Sensitivity for Execution Thoughts
> | **K Precision** | **Accuracy (\%)** | **V Precision** | **Accuracy (\%)** |
> |----------------|------------------|------------------|------------------|
> | 1              | 0                | 1                | 0                |
> | 2              | 21.5             | 2                | 27.6             |
> | 3              | 39.6             | 3                | 48.5             |
> | 4              | 52.94            | 4                | 52.94            |
> | 5              | 52.94            | 5                | 52.94            |
> | 6              | 52.94            | 6                | 52.94            |
> | 7              | 52.94            | 7                | 52.94            |
> | 8              | 52.94            | 8                | 52.94            |
>
> **Table D:** Quantization Sensitivity for Transition Thoughts
> | **K Precision** | **Accuracy (\%)** | **V Precision** | **Accuracy (\%)** |
> |----------------|------------------|------------------|------------------|
> | 1              | 10.5             | 1                | 11.9             |
> | 2              | 51.8             | 2                | 52.94            |
> | 3              | 52.94            | 3                | 52.94            |
> | 4              | 52.94            | 4                | 52.94            |
> | 5              | 52.94            | 5                | 52.94            |
> | 6              | 52.94            | 6                | 52.94            |
> | 7              | 52.94            | 7                | 52.94            |
> | 8              | 52.94            | 8                | 52.94            |
>
> The above analysis reveals that both the K and V caches of Transition thoughts are highly robust to aggressive quantization, substantiating our choice of 2-bit precision for T thoughts. Execution thoughts similarly tolerate 4-bit quantization with negligible accuracy loss. In contrast, the K cache for Reasoning thoughts exhibits noticeably higher sensitivity—mirroring the K/V precision asymmetry reported in QAQ—while the V cache remains relatively resilient. These findings validate the precision assignments used in ThinKV. Looking ahead, extending this analysis to incorporate outlier-handling techniques or mixed K and V precisions may further reduce accuracy loss under extreme compression regimes.
>
> In Figure 11(b) of the manuscript, we have presented a complementary ablation examining ThinKV’s precision assignment across two different models on the AIME and LiveCodeBench benchmarks. In this experiment, we employ the original formats used in the main paper—FP8 for 8-bit, NVFP4 for 4-bit, and ternary for 2-bit—which further corroborate the robustness trends observed above.
>
> ---

---

> ### Author Response · Authors · 2025-11-20
> **Author Response #3 to Reviewer 86j3**
>
> ## **W1: Fine-grained latency breakdown**
>
> Thank you for highlighting the importance of detailed latency analysis. Please see Fig. 1(c) for TPOT comparison of different methods. Table 5 in the main manuscript provides a per-operation time breakdown for ThinKV at batch size 1. Crucially, Table 5 also reports the percentage of decode steps that actually invoke thought refresh or eviction, showing that ThinKV’s proactive, coarse-grained eviction strategy (Fig. 10(b)) and periodic thought refresh allow it to operate overhead-free on more than 95\% of decode steps. This is in stark contrast to R-KV, its eviction and gather operations emerge as a major bottleneck (32.91\%) since it is invoked in nearly every decoding step.
>
> To further address your suggestion, we extend this analysis to varying batch sizes (8 and 32) and measure the relative time contribution of key operations (Attention, MLP, TBE Eviction, and Thought Refresh). For this analysis, we focus on a representative decode step that includes all mechanisms in action. Nevertheless, the rate of calls as presented in Table 4 remain largely identical across batch sizes. We present this analysis as a figure in Appendix E.14 of the revised manuscript. Our measurements show that ThinKV’s overhead (TBE eviction + thought refresh) remains minimal across batch sizes, consistently accounting for only ~14\% of the total latency, while Attention and MLP operations dominate with more than 80–85\% of the runtime. As batch size increases, the proportion of time spent in core model execution (attention, MLP) grows, confirming that ThinKV scales efficiently with increasing batch size.
>
>
> ---
>
> ## **W2: Extension to long input-context workloads**
>
> As noted in line 372 of the manuscript, we treat prefill tokens as R-type thoughts, i.e., they follow the same 4-bit precision assignment as decode-time R tokens. However, we do not evict from the prefill segment. This is because the observations in Section 3.3 applies specifically to decode-stage tokens, where attention sparsity and associations between thought types emerge naturally as the CoT unfolds. In contrast, prefill tokens constitute the entire grounding context for the task, and decode tokens continue to periodically attend back to them throughout the CoT. Evicting prefill tokens would therefore violate the core principle underlying TBE—namely, that eviction should target tokens of thought types with diminishing future attention utility.
>
> Applying a TBE-style eviction policy in addition to TBQ (ThinKV=TBQ+TBE) to prefill tokens requires further study. We note this limitation in Appendix F and leave it as future work. **Nonetheless, applying TBQ alone to prefill tokens still yields a 4× compression of the prefill KV cache, ensuring meaningful memory savings even without eviction**.
>
> ThinKV, however, is orthogonal to other prefill-optimization techniques, and can be combined with them seamlessly. To illustrate this, we additionally evaluate ThinKV integrated with SnapKV on LongBench v2 [1]. LongBench v2 is a suite of long-context evaluation tasks characterized by both large input contexts and substantial output lengths. For this study, we evaluate on R1-LLama-8B and select the Code Repo QA (along the lines of SWE-Bench) task from LongBench v2, which features 16K–32K average input lengths and output sequences up to 8K tokens, making it a representative long-input/long-output setting. Prefill tokens are quantized to 4 bits using TBQ and evicted according to SnapKV’s eviction policy under varying budgets, while the decode stage employs ThinKV with a fixed KV-cache budget of 512 tokens. Our results are summarized in Table E below.
>
> **Table E:** Ablation of Prefill and Decode Settings for Hybrid (SnapKV +ThinKV) Compression
>
> | **Method**        | **Prefill Precision/Eviction Budget** | **Decode Precision/Eviction Budget** | **Accuracy** |
> |-------------------|--------------------------|--------------------------|--------------|
> | FullKV            | –                        | –                        | 30           |
> | ThinKV            | 4-bits                   | 3.8/512                 | 28           |
> | SnapKV + ThinKV   | 4-bits/2048              | 3.7/512                 | 24           |
> | SnapKV + ThinKV   | 4-bits/4096             | 3.8/512                 | 28           |
>
> Evidently, ThinKV incurs only a small accuracy drop, despite achieving 4x prefill compression and up to 66x decode compression. When combined with SnapKV, accuracy depends largely on the prefill eviction budget: 2048-token budget reduces accuracy by 6\%, while a 4096-token budget yields only a modest accuracy degradation.
>
> ---
>
> We hope our clarifications resolve any apprehensions and contribute to an even more favorable assessment of our work, and we would be glad to address any follow-up questions the reviewer may have.

---

> ### Comment · Reviewer_86j3 · 2025-11-20
> **Acknowledge for Authors Reply.**
>
> * **Q1**: Thanks for the authors experments on hybrid reasoning models. The reply well adressed my concerns. Proposed ThinkKV works will on these models.
> * **Q2**: The sensitivity analysis shows well that transistion thoughts are less sensitive to quantization. (A typo in Appendix E.9, the refered figure should be Figure 16 rather than Figure 9.)

---

> > ### Author Response · Authors · 2025-11-20
> > **Author Acknowledgement**
> >
> > Thank you for your quick response and for letting us know that our rebuttal fully addressed your concerns. We’re glad to hear this, and we also appreciate you catching the typo — it has been corrected.

---

> ### Author Response · Authors · 2025-11-28
> **[ThinKV] Follow-Up on Remaining Concerns Before Discussion Ends**
>
> Dear Reviewer 86j3,
>
> As we approach the end of the discussion period, we would like to check whether you have any additional requests for experiments or clarifications. We are happy to provide further analyses if needed, to earn an even more positive reconsideration of our work. If our current rebuttal and revisions address your concerns, we would be grateful if you would consider updating your score accordingly to reflect the positive post-rebuttal assessment of our work, should you find it appropriate.
> Thank you again for your constructive feedback.
>
> Regards
>
> Authors of ThinKV

---

### Official Review · Reviewer_pvM3 · 2025-10-30

**Soundness:** 3
**Presentation:** 3
**Contribution:** 3
**Rating:** 6
**Confidence:** 3

**Summary:**

This paper presents a combination of strategies to reduce the KV cache size during inference with a decoder transformer.

The method combines quantization and deletion of blocks of tokens in the KV cache based on a category which is inferred from the attention sparsity. The authors also propose an efficient implementation derived from PageAttention.

The resulting gain in speed for a given accuracy level beats existing methods.

**Strengths:**

The problem addressed by the authors is key in the deployment of large language models, and more generally reducing memory footprint is a central issue: Lower memory means possibly larger batches, larger model dimension, longer contexts, and longer / more c-o-t, which ultimately translate into smarter models.

**Weaknesses:**

This is an highly engineering paper. The method proposed does not derive from a new principle nor rely on formal results or quantitative analysis. There are some motivating experiments that are interesting, but the bulk of the method is a recipe with lot of hyperparameters.

For the presentation, I encourage the authors to change the scaling of fig 1 and have both axes starting from 0, the current framing is IMO misleading.

I am doubtful the inclusions of figures respect ICLR's style, and captions could be more detailed (e.g. fig 3, what data set?)

**Questions:**

Some questions, in no specific order:

- The clear separation of the three classes is very impressive in Fig 3, and Fig 13 provides other examples, for different models, but how does it hold on other datasets? Are all this graphs from AIME?

- There are many hyperparameters in 3.2 and 3.3, how were they chosen?

- The method is compared to other KV cache quantization or eviction, but what are the sota methods overall? What about other types of methods (MLA, YOCO)? It would be great to give a clearer sense of the pareto front accuracy vs. memory footprint across methods.

---

> ### Author Response · Authors · 2025-11-20
> **Author Response #1 to Reviewer pvM3**
>
> We sincerely thank the reviewer for taking the time to provide a constructive evaluation of our work. We greatly appreciate the recognition of our strong empirical results. We address the reviewer’s questions and concerns in detail below. For clarity, we mark responses to weakness comments with **W#** and responses to explicit questions with **Q#**.
>
> ---
>
> ## **W1: ThinKV is pragmatically motivated and is not based on strong theoretical and quantitative analysis.**
>
> We appreciate the reviewer’s perspective and have revised the manuscript to explicitly clarify the underlying principles in Section 3. Specifically, ThinKV is the first to identify and leverage the *degree of attention sparsity* to systematically differentiate thought types. This decomposition arises from quantitative analysis of layer-wise attention sparsity trajectories across thousands of decoded reasoning traces (Section 3.1; Figure 3), and is formalized in Definition 1 of the manuscript. To further validate this categorization, we associate interpretable, representative keywords for each thought segment (Appendix D.2), demonstrating consistency between the sparsity-driven decomposition and distinct semantic behaviors observed in LRMs. This principled thought-type differentiation motivates ThinKV’s hybrid quantization–eviction strategy and distinguishes it from prior token-level heuristics.
>
> Section 2 analytically characterizes how different forms of compression affect the KV-cache footprint and demonstrates empirically (Fig. 2) why quantization-only or eviction-only strategies cannot maintain a stable compression–accuracy trade-off. This motivates the unified hybrid method introduced in ThinKV. The hyperparameters (design choices) involved are a direct instantiation of this analysis and are examined through extensive ablations in Section 6.3, where we show that each contributes meaningfully to the overall behavior.
>
> Finally, the Continuous Thinking kernel (Section 5.2) follows from a formal analysis of gather overhead—a previously overlooked bottleneck in decode-time KV eviction—and is included to ensure end-to-end system efficiency.
>
> ---
>
> ## **W2: Update figures and captions.**
>
> Thank you for catching this typo — we have corrected Figure 1(c) in the revised manuscript. We have also updated all figure captions to be more descriptive and improve readability.
>
> ---
>
> ## **Q1: The clear separation of the three classes is very impressive in Fig 3, and Fig 14 provides other examples, for different models, but how does it hold on other datasets ?**
>
> Thank you for recognizing the strengths of our attention-sparsity guided thought separation. Figure 14 includes examples from other datasets, including LiveCodeBench. In the revised manuscript, we have updated Figure 3 to present samples from two datasets and two models to better illustrate the generality of the three-class separation. We have also added explicit dataset labels to all plots in Figure 14 to make this correspondence clear.
>
> ---
>
> ## **Q2: How are the hyperparameters chosen ?**
>
> We thank the reviewer for raising this important question. We emphasize that **ThinKV does not require per-model or per-dataset tuning**; a single well-chosen configuration works robustly across all evaluated models and benchmarks. The hyperparameters in Sections 4.1–4.3 (revised) are design choices based on empirically grounded ablation process.
>
> For **thought decomposition** (Section 4.1), the number of thought types \(|T|\), refresh interval \(\tau\), and optimal layer subset \(|L^*|\) are ablated in Fig. 11(a) and Fig. 10(c) and discussed in Section 6.3.
>
> For **TBQ** (Section 4.2), the quantization group size \(g = 16\) follows established practice (Alvarez et al., 2025; NVFP4). We adopt 2/4/8-bit levels because they are hardware-aligned and enable mapping precision to thought importance; this mapping is validated in Fig. 11(b) and analyzed in Section 6.3. Appendix E.8 further confirms that NVFP4, FP8, and ternary formats provide the most effective trade-offs. Furthermore, we also conducted a quantization sensitivity analysis in Appendix E.9 and find this to further validate the precision assignments used in ThinKV.
>
> For **TBE** (Section 4.3), the retention schedule \(R = \{64, 32, 16, 8, 4\}\) and minimum retention of 4 tokens were selected based on ablations (Fig. 10a). We use powers-of-two retention to ensure memory alignment, following RocketKV (Behnam et al., 2025) and MicroScopiQ (Ramachandran et al., 2025). The progressive annealing (Fig. 10(b)) schedule preserves essential contextual traces and avoids abrupt loss of reasoning continuity.
>
> Overall, these design choices **directly operationalize our motivating analysis** and are thoroughly ablated in Section 6.3 and Appendix E.
>
> ---

---

> ### Author Response · Authors · 2025-11-20
> **Author Response #2 to Reviewer pvM3**
>
> ## **Q3: What are the SoTA methods overall and what about MLA and YOCO ?**
>
> Thank you for this valuable question. In the setting of training-free, decode-time KV-cache compression, current state-of-the-art methods fall into two main categories: eviction-based and quantization-based approaches. For eviction, R-KV (NeurIPS 2025) is the strongest published baseline, with LazyEviction and RaaS as the next most competitive methods. For quantization, PM-KVQ currently provides the strongest results. ThinKV is evaluated against all of these representative SoTA baselines in our experiments.
>
> MLA and YOCO introduce fundamentally new model architectures, placing them outside the regime of training-free, plug-in KV-cache compression methods such as ThinKV. For this reason, direct comparison is not strictly fair, though ThinKV can in principle be integrated with any model architecture, including MLA-style designs. We note that MLA models are not yet widely available and we designate it as important future work. Consistent with prior work, our evaluations therefore use the R1-distilled models alongside several other model families.
>
> ---
>
> ## **Q4: What is the Pareto front of accuracy v/s memory footprint across methods ?**
>
> Thank you for suggesting this. We now include an Accuracy vs. KV cache size plot in Appendix E.11, comparing ThinKV against SoTA baselines across varying eviction budgets and KV-cache precisions. We conclusively demonstrate that ThinKV sets a new Pareto-frontier.
>
> ---
>
> We hope our clarifications resolve any apprehensions and contribute to an even more favorable assessment of our work, and we would be glad to address any follow-up questions the reviewer may have.

---

### Official Review · Reviewer_ESb6 · 2025-10-31

**Soundness:** 2
**Presentation:** 1
**Contribution:** 2
**Rating:** 4
**Confidence:** 4

**Summary:**

The paper proposes a hybrid quantization+eviction methods for reasoning model, achieve high compression ratio with minimum accuracy degradation.

**Strengths:**

1. The paper combines eviction with quantization, and shows that different tokens should quantized differently.
2. Experiments are run on latest reasoning models and challenging benchmarks.

**Weaknesses:**

1. A very unfortunate thing for post-training efficiency technique of reasoning models is the inflation of generation length. Quantization is servere as mentioned in the paper. Sparsity attention can also lead to inflation but tends to be smaller. So it is meaningless to show metrics like **throughputs** or **compression ratio** in the paper as they hide this effect. Similar to max-batch size. You will never get these high batch size as 1. larger batch size comes from larger compression ratio-> 2. larger compression ratio means more inflation of kv cache-> 3. larger kv cache per seq means lower batch size. **So from my side, the only reasonable metric for reasoning efficiency evaluation is how long you take to solve an benchmark e2e such as AIME and LCB using same GPU setting.** And this should be normalized to standard serving framework with dynamic batching like vllm or sglang so that it reflects the ablility of your kv compression method.
2. The paper writing is not clear. It's difficult to follow. I suggest the methodology just focuses on describing what the algorithm looks like without so many definition, observation and notation. Even with so many paragraphs, I have not been convinced and motivated. For example, what if future reasoning model reasons in hidden states instead of explicit outputing tokens? Is there still going to be R, T, E? I am not sure, but I believe attention sparsity and quantization still exisit.

**Questions:**

1. How many times of averaged score do you use for one reasoning question?
2. Can you explain more on block_size in flg 9e?

---

> ### Author Response · Authors · 2025-11-20
> **Author Response #1 to Reviewer ESb6**
>
> We sincerely thank the reviewer for the thoughtful and constructive evaluation of our work. For clarity, we mark responses to weakness comments with **W#** and responses to explicit questions with **Q#**.
>
> ---
>
> ## **W1: Shouldn’t reasoning model efficiency be evaluated based on end-to-end latency and normalized to serving frameworks such as vLLM?**
>
> We fully agree with the reviewer’s recommendation, and we revised our manuscript to adopt end-to-end latency as another evaluation metric. We have been actively working on integrating ThinKV into vLLM and conducted two complementary evaluations:
> (i) a dynamic serving analysis under multi-user concurrency [2], measuring real system throughput (Reqs/s) vs. E2E User latency (s), and (ii) a static analysis with batch size 1, focusing on time-per-request (TPR) and Intelligence/Watt [1] metrics. All experiments have been conducted on a single A100-80GB GPU.
>
> #### (i) Dynamic Serving Analysis (ThinKV achieves 38\% higher loads (Reqs/s), 27\% lower latency than R-KV.)
>
> **Table A:** Comparison of system throughput (requests per second) and end-to-end user latency across FullKV, R-KV, and ThinKV for increasing batch sizes in vLLM.
>
> | **Batch Size** | **FullKV Throughput (Req/s)** | **FullKV User Latency (s)** | **R-KV Throughput (Req/s)** | **R-KV User Latency (s)** | **ThinKV Throughput (Req/s)** | **ThinKV User Latency (s)** |
> |----------------|-------------------------------|------------------------------|------------------------------|-----------------------------|-------------------------------|------------------------------|
> | 1              | 0.005                         | 195                          | 0.005                        | 188                         | 0.005                         | 184                          |
> | 2              | 0.009                         | 215                          | 0.011                        | 188                         | 0.011                         | 186                          |
> | 4              | 0.013                         | 301.5                        | 0.020                        | 196                         | 0.021                         | 193                          |
> | 8              | 0.017                         | 480                          | 0.039                        | 204                         | 0.040                         | 202                          |
> | 16             | OOM                             | OOM                            | 0.073                        | 219                         | 0.077                         | 208                          |
> | 32             | OOM                             | OOM                            | 0.136                        | 236                         | 0.142                         | 226                          |
> | 64             | OOM                             | OOM                            | 0.163                        | 392                         | 0.215                         | 298                          |
> | 128            | OOM                             | OOM                            | 0.226                        | 566                         | 0.275                         | 465                          |
> | 256            | OOM                             | OOM                            | 0.259                        | 989                         | 0.356                         | 719                          |
>
> Motivated by the dynamic-serving analyses in [2,3], we evaluate ThinKV under multi-user concurrency. For a batch size of B, we inject B parallel requests to emulate B active users and measure the achieved real system throughput (requests/s) together with the average E2E latency experienced by each user. If the method cannot support a batch size larger than a given B without running OOM, queuing delays inflate latency without increasing system throughput [2]; we therefore focus on the performance of requests that are actively executed (please let us know if you'd like to incorporate queuing delays).
>
> Table A summarizes our findings, all reported measurements are averaged across 3 runs with B requests randomly sampled from AIME dataset with a cache budget of 1024. We added this table as Figure 9 of Section 6.2 in the revised manuscript. FullKV cannot sustain batch size more than B=8. Under an iso-batch comparison with FullKV at B=8, ThinKV achieves up to 58\% lower latency while satisfying higher requests per second. ThinKV is able to support larger batch sizes (see Table 2 in manuscript) due to high compression of the KV cache. At B=256, under iso-batch conditions, ThinKV achieves 38\% higher loads (Reqs/s) and 27\% lower latency compared to R-KV. This demonstrates that ThinKV not only improves per-request efficiency but also scales more gracefully under heavy concurrency, making it a more robust choice for practical serving workloads.

---

> ### Author Response · Authors · 2025-11-20
> **Author Response #2 to Reviewer ESb6**
>
> #### (ii) Static analysis with a batch size of 1 (ThinKV reduces E2E latency by up to 9\%)
>
> **Table B:** End-to-end average reasoning latency (Time-per-request (TPR)) at batch size 1, accuracy, and Intelligence/Watt on AIME dataset for R1-Llama-8B under varying token budgets, comparing FullKV, R-KV (seq/ovl), and ThinKV. All measurements conducted on a single A100-80GB GPU.
>
> | **Method**      | **Token Budget** | **TPR (s)** | **Accuracy (\%)** | **Intelligence/Watt** |
> |-----------------|------------------|-------------|-------------------|------------------------|
> | FullKV          | –                | 259.6       | 50.0              | 0.20                   |
> | R-KV (seq)      | 512              | 242.6       | 40.0              | 0.17                   |
> | R-KV (ovl)      | 512              | 240.8       | 40.0              | 0.17                   |
> | ThinKV          | 512              | 237.5       | 46.7              | 0.21                   |
> |-----------------|------------------|-------------|-------------------|------------------------|
> | R-KV (seq)      | 1024             | 247.8       | 46.7              | 0.20                   |
> | R-KV (ovl)      | 1024             | 246.0       | 46.7              | 0.20                   |
> | ThinKV          | 1024             | 243.6       | 50.0              | 0.22                   |
> |-----------------|------------------|-------------|-------------------|------------------------|
> | R-KV (seq)      | 2048             | 254.2       | 50.0              | 0.20                   |
> | R-KV (ovl)      | 2048             | 253.7       | 50.0              | 0.20                   |
> | ThinKV          | 2048             | 251.0       | 50.0              | 0.21                   |
>
> Table B reports the average end-to-end request latency (Time-per-Request, TPR), accuracy, and Intelligence/Watt [1] for various KV-compression strategies evaluated on the AIME benchmark using R1-Llama-8B. ThinKV at a token budget of 1024, while simultaneously achieving lossless compression, is able to achieve up to 6\% lower latency on average per request as compared to the FullKV baseline. These gains extend beyond what a highly optimized framework like vLLM already provides, and ThinKV’s benefits become especially pronounced at larger batch sizes (refer Table A).
>
> Recent works have demonstrated that Intelligence/Watt [1]  offers a unified view of both capability and efficiency, making it a principled metric for comparing compression strategies. As shown in Table B, these latency improvements materially increase ThinKV’s Intelligence/Watt over FullKV and R-KV. We have updated the revised manuscript with this information in Appendix E.15.
>
>
> #### Clarifications on Generation Length
>
> We have revised Fig. 10(d) in the paper to include the impact of TBE and ThinKV (TBQ+TBE) on the generation length. On tasks such as LiveCodeBench, ThinKV increases generation length by 1.28×, and this overhead is outweighed by the benefits of KV-cache compression—namely reduced attention computation, lower memory footprint (5\% of the FullKV), and the ability to support substantially higher batch sizes (up to 54x).
>
> [1] Saad-Falcon, J., Narayan, A., Akengin, H. O., Griffin, J., Shandilya, H., Lafuente, A. G., ... & Ré, C. (2025). Intelligence per Watt: Measuring Intelligence Efficiency of Local AI. arXiv preprint arXiv:2511.07885.
>
> [2] Yu, G. I., Jeong, J. S., Kim, G. W., Kim, S., Chun, B. G. (2022). Orca: A distributed serving system for {Transformer-Based} generative models. In 16th USENIX Symposium on Operating Systems Design and Implementation (OSDI 22) (pp. 521-538).
>
> [3] Rehg, Isaac. "Kv-compress: Paged kv-cache compression with variable compression rates per attention head." arXiv preprint arXiv:2410.00161 (2024).
>
> ---
>
> ## **W2: Suggestion to improve paper writing and restructure to separate motivation and methodology section.**
>
> In the revised manuscript, we have restructured the motivation and methodology sections to create a clearer separation between the empirical observations that motivate ThinKV and the algorithmic design that follows from them.
>
> ---

---

> ### Author Response · Authors · 2025-11-20
> **Author Response #3 to Reviewer ESb6**
>
> ## **Q1: What if future reasoning model reasons in hidden states instead of explicitly outputting tokens?**
>
> We clarify that our method does not depend on the model explicitly emitting tokens to distinguish thought types. In contrast to prior work that infers thought structure from surface-form tokens [1], ThinKV is built on a principled, sparsity-driven formulation. Thought types are differentiated through attention sparsity patterns, derived from quantitative analysis of layer-wise sparsity trajectories across thousands of traces.
>
> Moreover, our formulation does not assume the existence of exactly three thought categories. As discussed in Sections 3.1 and 4.1, the cardinality ∣T∣ is a design choice that enables the algorithm to identify the number of thought types best supported by the underlying sparsity structure. Through ablations shown in Figure 11(a) and Section 6.3, we consistently found ∣T∣ = 3 to be optimal for all evaluated models. As long as future models continue to rely on attention-based architectures—which currently remain the most effective and widely adopted paradigm—ThinKV remains directly applicable.
>
> In a theoretical setting where future models depart from attention-based architectures, ThinKV could still be adapted by incorporating steering vectors [1,2]. Steering vectors operate directly in the latent space and, with sufficient exploration, may offer an effective alternative to attention-sparsity–based thought differentiation if such a mechanism becomes necessary.
>
> [1] Venhoff, Constantin, et al. "Understanding reasoning in thinking language models via steering vectors." arXiv preprint arXiv:2506.18167 (2025).
>
> [2] Huang, Yao, et al. "Mitigating Overthinking in Large Reasoning Models via Manifold Steering." arXiv preprint arXiv:2505.22411 (2025).
>
> ---
>
> ## **Q2: How many times of averaged score do you use for one reasoning question?**
>
> Thank you for the question. We follow the evaluation protocol used in R-KV and adopt **pass@1** as our accuracy metric. For each question, we generate **8 independent responses** and compute pass@1 as $\text{pass@1} = \frac{1}{k} \sum_{i=1}^{k} p_i$, where $p_i$ denotes whether the $i$-th sampled response is correct. Similarly for all performance measurements, throughput and latency numbers are obtained by averaging across 3 independent runs. We have updated Section 6.1 of the revised manuscript to reflect this.
>
> ---
>
> ## **Q3: Can you explain more on block size in Fig 10e (revised)?**
>
> In ThinKV and vLLM, block\_size refers to the number of consecutive KV-cache tokens that are grouped and stored together as a single memory block within the paged KV-cache. vLLM’s PagedAttention organizes the KV cache of each sequence into fixed-size blocks of size block\_size to enable efficient non-contiguous memory allocation and reuse, and ThinKV builds directly on this design (see Figure 6, where A,B, C and D are tokens within a block of block\_size=4). The choice of block size can have a substantial impact on the performance [1]. If the block size is too small, ThinKV integrated into vLLM may not fully utilize the GPU’s parallelism for reading and processing KV cache. If the block size is too large, internal fragmentation increases, and each block may pack more thought segments per block, incurring substantial metadata overhead in the block table and increasing eviction time, which degrades performance.
>
> Figure 10(e) presents our ablation over different ThinKV block sizes to determine the optimal configuration. From this analysis, block sizes of 8 and 16 tokens offer the strongest performance. We adopt a block size of 8 tokens, as it minimizes metadata overhead—fewer thought segments are forced into the same block. Notably, this choice aligns directly with vLLM’s own block-size analysis (Figure 18(b) in [1]).
>
> [1] Kwon, W., Li, Z., Zhuang, S., Sheng, Y., Zheng, L., Yu, C. H., ... \& Stoica, I. (2023, October). Efficient memory management for large language model serving with pagedattention. In Proceedings of the 29th symposium on operating systems principles (pp. 611-626).
>
> ---
>
> We hope our clarifications resolve any apprehensions and contribute to an even more favorable assessment of our work, and we would be glad to address any follow-up questions the reviewer may have.

---

> > ### Comment · Reviewer_ESb6 · 2025-11-26
> >
> > Thanks for the detailed rebuttal. It resolves all my previous concerns. I just have a minor question that whether the evicted KV cache can somewhat be retrieved back in the future or it's lost forever.

---

> ### Author Response · Authors · 2025-11-26
> **Author Acknowledgement to Reviewer ESb6**
>
> We are truly delighted to hear that our rebuttal has addressed all of the reviewer’s previous concerns. Thank you very much for taking the time to carefully read our responses and for engaging so thoughtfully with our work—we greatly appreciate it.
>
> **Answer to follow-up question:** Consistent with prior work on KV cache compression for LLMs and LRMs, evicted KV entries are discarded and cannot be retrieved (the token-dropping paradigm). However, as our experiments in Figure 10(a) demonstrate, ThinKV sustains recall rates close to FullKV across a wide range of token budgets. Recall rate measures the fraction of important tokens (Top-10) preserved by a compression method relative to those under full attention at each decoding step. Consequently, ThinKV’s thought-adaptive eviction scheme obfuscates the need to retrieve evicted tokens, as it consistently retains the optimal subset of high-impact tokens. This behavior is further corroborated by our near-lossless performance across models and datasets even at a 1024-token KV budget.
>
> That said, should retrieval of evicted tokens ever be required, the techniques proposed in ThinKV are orthogonal to the token offloading and retrieval literature [1,2]. Along these lines, after ThinKV’s TBE stage, evicted tokens could be offloaded to CPU memory and later retrieved for subsequent decoding steps. **We would be happy to conduct such an experiment if the reviewer finds it valuable**.
>
> References:
>
> [1] Sheng, Ying, et al. "Flexgen: High-throughput generative inference of large language models with a single gpu." International Conference on Machine Learning. PMLR, 2023.
>
> [2] Hao, Jitai, et al. "Omnikv: Dynamic context selection for efficient long-context llms." The Thirteenth International Conference on Learning Representations. 2025.

---

### Official Review · Reviewer_P5bA · 2025-11-02

**Soundness:** 4
**Presentation:** 4
**Contribution:** 3
**Rating:** 8
**Confidence:** 3

**Summary:**

ThinKV is a KV-cache compression framework for large reasoning models (LRMs) that exploits attention sparsity to decompose chain-of-thought (CoT) into three semantic segments: reasoning (R), execution (E), and transition (T). It applies hybrid compression, including quantization, eviction and system co-design. Across DeepSeek-R1, GPT-OSS, AceReason (math/code benchmarks), ThinKV achieves <5% KV-cache with near-lossless accuracy and 5.8× throughput against previous SOTA methods.

**Strengths:**

- The overall quality of this paper is pretty good. The authors demonstrate deep insights in their observation and analysis of the problem and propose an innovative algorithmic design and operator implementation.

- The overall presentation and expression are excellent.

- The paper demonstrates strong empirical results. It sustains >95% accuracy with 20–40× compression on AIME/LiveCodeBench, while 3× batch size and 5.8× throughput gains over R-KV/H2O.

**Weaknesses:**

1. The current experimental setup shows a relatively large accuracy drop. It is recommended that the authors investigate how much acceleration can still be achieved when the accuracy drop is controlled within 1%, to better demonstrate the stability and practical value of the proposed method.

2. The paper appears to lack **ablation studies** related to the operator design. The authors are encouraged to add relevant experiments to analyze the impact of different operator designs and usages on overall throughput and accuracy, thereby enhancing the completeness and persuasiveness of the paper.

**Questions:**

1. The experimental results show that the proposed method achieves a significant speedup. However, according to Appendix E.2, the implementation is based on **Transformers**. It is suggested that the authors further evaluate the throughput performance on industrial deployment frameworks such as **vLLM** or **SGLang**, to verify the acceleration effect in practical scenarios and compare it with other methods (e.g., the **Double Sparse** method in **SGLang**).

2. It is suggested to include additional KV cache compression methods beyond **H2O** and **RKV** in the experiments, so as to better demonstrate the superiority of the proposed approach.

---

> ### Author Response · Authors · 2025-11-20
> **Author Response #1 to Reviewer P5bA**
>
> We sincerely thank the reviewer for taking the time to provide such a thoughtful and encouraging evaluation of our work. We greatly appreciate the recognition of our writing and presentation, contributions and strong empirical results. We address the reviewer’s questions and concerns in detail below. For clarity, we mark responses to weakness comments with **W#** and responses to explicit questions with **Q#**.
>
> ---
>
> ## **W1: How much acceleration can be achieved when the accuracy drop is limited to within 1\%?**
>
> We appreciate the reviewer’s question. In the paper, we primarily report performance results (Table 2) with a 1024-token budget and R4E4T2 precision assignment (default) because this setting satisfies the ≤1\% accuracy drop criterion for the majority of evaluated LRMs and datasets (e.g., R1-Llama-8B, R1-Llama-70B on AIME and MATH-500).
>
> **However**, we observed that a small subset of models—particularly GPT-OSS-20B on AIME and LiveCodeBench—can exhibit an accuracy drop slightly above 1\% at this setting. To provide a more conservative, accuracy-preserving configuration, we additionally report results with a 2048-token budget combined with R4E4T2 quantization. Under this configuration, all evaluated models on all datasets consistently remain within a <1\% accuracy drop, while still benefiting from ThinKV’s high compression and throughput. We summarize these results in Table A below and have incorporated it as Table 3 in Section 6.2 in the revised manuscript.
>
> **Table A:** ThinKV throughput (tokens/s) for R1-Llama-8B on an A100-80GB GPU using a 2048-token budget performing continuous generation of 32K tokens on AIME prompts. Batch size reflects the maximum achievable value before OOM.
> | **Method** | **Accuracy** | **Batch Size (max)** | **Token Budget** | **Throughput** |
> |------------|--------------|-----------------------|-------------------|-----------------|
> | FullKV     | 50           | 13                    | –                 | 297.5           |
> | **ThinKV** | 50           | **290**               | **2048**          | **4688.4**      |
>
> Evident from Table A above, ThinKV, while sustaining a ≤1\% accuracy drop relative to FullKV, enables a drastic increase in usable batch size (13 → 290) and delivers a 15.8× throughput improvement. This demonstrates that even under conservative, accuracy-preserving settings, ThinKV enables substantial system-level acceleration without compromising model quality.

---

> ### Author Response · Authors · 2025-11-20
> **Author Response #2 to Reviewer P5bA**
>
> ---
>
> ## **W2: Ablations on operator (i.e., TBQ, TBE) design.**
> In Section 6.3 (Table 4 of the revised manuscript), we originally presented ablations that individually quantify the accuracy contribution of ThinKV’s core components i.e., TBQ and TBE. Following your suggestion, we have updated Table 4 (shown as Table B below) with the throughput and latency impact of ThinKV's components while also simultaneously sweeping different token budgets for TBE (without interference from TBQ) to provide an operator-isolated assessment. We also direct the reviewer to Figure 11(b) in the revised manuscript, which ablates the effect of assigning different precisions to R, T, and E thoughts. Moreover, we now also include a detailed quantization-sensitivity analysis of keys and values across different thought types in Appendix E.9.
>
> **Table B:** Accuracy, throughput, and latency (iso-batch) impact of ThinKV’s components, evaluated on GPT-OSS-20B using the LiveCodeBench benchmark.
> | **Method**              | **Avg. Precision / Eviction Budget** | **Accuracy** | **Batch Size**   | **Norm. Throughput**   | **Norm. Latency**  |
> |-------------------------|----------------------------------------|--------------|----------------|------------------------|--------------------|
> | FullKV                  | –                                      | 77.8         | 8              | 1x                      | 1x                  |
> | TBQ                     | 3.5                                    | 77.8         | 8              | 1.1x                    |  0.98x           |
> | TBE                     | 512                                    | 62.5         | 8              | 1.78x                   |  0.36x            |
> | TBE                     | 1024                                   | 76.9         | 8              | 1.48x                   |  0.38x             |
> | TBE                     | 2048                                   | 77.8         | 8              | 1.27x                   | 0.44x            |
> | **ThinKV (TBQ+TBE)**    | **3.8, 1024**                          | **76.4**     | **8**          |  **1.51x**                 | **0.42x**             |
>
> From Table B, we observe that ThinKV (TBQ+TBE) delivers clear throughput and latency gains even under an iso-batch comparison, while incurring only a marginal accuracy reduction relative to FullKV. More importantly, these iso-batch results understate ThinKV’s full benefit: because the hybrid compression (TBQ+TBE) substantially compresses the KV cache footprint (requiring <3.67\% of FullKV memory), ThinKV can sustain much larger batch sizes than the baseline, leading to substantial end-to-end performance improvements as shown in Table 2 in the revised manuscript.
>
> ---
>
> ## **Q1: Can the authors evaluate the throughput performance of the proposed method on industrial deployment frameworks such as vLLM or SGLang ?**
>
> Thank you for raising this important point. We have been actively working on integrating ThinKV in vLLM. Please see Appendix E.2 for details on our vLLM implementation. As shown in Table C below, we report ThinKV’s throughput under two iso-batch comparisons: (i) batch size = 8 against FullKV and R-KV (ovl), and (ii) batch size = 256 against R-KV (ovl). We have also updated our manuscript with these results in Appendix E.12.
>
> **Table C:** Throughput (tokens/sec) evaluation of R1-Distill-Llama-8B during continuous 32K-token generation on an A100-80GB GPU, comparing FullKV, R-KV (ovl), and ThinKV integrated into vLLM.
>
> | **Method**      | **Batch Size** | **Budget** | **Throughput ** |
> |-----------------|----------------|------------|----------------|
> | FullKV          | 8              | –          | 228.5          |
> | R-KV (ovl)      | 8              | 1024       | 331.9          |
> | ThinKV          | 8              | 1024       | 346.9          |
> |-----------------|----------------|------------|----------------|
> | R-KV (ovl)      | 256            | 1024       | 4883.3         |
> | ThinKV          | 256            | 1024       | 6622.4         |
>
> Furthermore, in Figure 9 we present system throughput (reqs/sec) vs. e2e user latency (sec), also evaluated within the vLLM framework.

---

> ### Author Response · Authors · 2025-11-20
> **Author Response #3 to Reviewer P5bA**
>
> ---
>
> ## **Q2: Including additional KV cache compression methods beyond H2O and RKV in the experiments**
>
> We would like to humbly clarify that our evaluations include a broad set of recent and state-of-the-art KV-cache compression baselines spanning both quantization and eviction approaches as described in Section 6.1. For eviction methods, we evaluate LazyEviction and RaaS in addition to H2O and R-KV, as shown in Figure 1 and 8. For quantization methods, we include KIVI and PM-KVQ, as summarized in Table 1. Across all these baselines, ThinKV consistently demonstrates superior accuracy under diverse compression ratios. For throughput evaluations, we focus primarily on R-KV because it is the closest state-of-the-art baseline to ThinKV in terms of accuracy, and it shares the gather-based eviction mechanism employed by many recent methods, including LazyEviction and RaaS. As such, R-KV serves as a strong and representative baseline for system-level comparison.
>
> ---
>
> We hope our clarifications resolve any apprehensions and contribute to an even more favorable assessment of our work, and we would be glad to address any follow-up questions the reviewer may have.

---

> ### Comment · Reviewer_P5bA · 2025-11-27
>
> I appreciate the authors' response, which address my questions. I would keep my score.

---

> > ### Author Response · Authors · 2025-11-27
> > **Author Acknowledgement**
> >
> > We truly appreciate the time and care you dedicated to reviewing our work, and we’re grateful for your positive comments. We’re pleased to hear that our rebuttal fully addressed your concerns, and we sincerely thank you for your constructive feedback throughout the process.

---

### Author Response · Authors · 2025-11-20
**General Response by Authors (Part 2 of 2)**

## Summarization of Rebuttal Comments and Revision

###  Reviewer `P5bA`
1. How much acceleration can be achieved when the accuracy drop is limited to within 1\%?
   - ThinKV, with a 2048-token budget and R4E4T2 quantization, keeps accuracy within 1% across all models and datasets while delivering a 15.8× throughput gain.
   - Updated Section 6.2 to include the newly added Table 3.
1. Ablations on operator (i.e., TBQ, TBE) design on accuracy and throughput?
   - Updated Table 4 to now also include throughput and latency measurements of each component of ThinKV at iso-batch (batch size = 8).
1. Throughput numbers of ThinKV integrated into vLLM
   - Outlined specifics of vLLM integration in Appendix E.2.
   -  Added Appendix E.12 and Table 12, demonstrating ThinKV achieving 1.35x higher throughput than R-KV (ovl) at batch size of 256.
1. Additional KV cache compression methods beyond H2O and RKV
   - Answer provided in response to reviewer `P5bA` Q2.
   - We benchmark contemporary eviction methods (LazyEviction, RaaS, H2O, R-KV) and quantization methods (KIVI, PM-KVQ).

###  Reviewer `ESb6`
1. End-to-end latency evaluations and integration into vLLM
   - We perform two key experiments: (i) Dynamic serving analysis under multi-user concurrency, measuring real system throughput (Reqs/s) vs. E2E User latency (s) and (ii) Static analysis with a batch size of 1, measuring Time-per-request (TPR) and Intelligence/Watt
   - At B=256, under iso-batch conditions, ThinKV achieves 38\% higher loads (Reqs/s) and 33\% lower latency compared to R-KV.
   - Added new Figure 9 and updated discussion in Section 6.2 and Appendix E.15.
   - Updated Figure 10(d) to include TBE and ThinKV's impact on generation length.
1. Separate motivation and algorithm section
   - Restructured paper to include a dedicated Section 3 for motivation and Section 4 and 5 for algorithm and system methodology respectively
1. Future Outlook of ThinKV
   - Answer provided in response to reviewer `ESb6` Q1.
1. Clarification on evaluation methodology
   - Answer provided in response to reviewer `ESb6` Q2.
   - Updated Section 6.1.
1. Clarification on block size
   - Answer provided in response to reviewer `ESb6` Q3.

###  Reviewer `pvM3`
1. Key contributions, analysis and central principle behind ThinKV
   - Answer provided in response to reviewer `pvM3` W1.
1. Update figures and captions
   - Updated axes of Figure 1( c ).
   - Updated all figure captions to be more descriptive.
1. Clear separation of thought types in Figure 3
   - Updated Figure 3 to include samples from LiveCodeBench as well and improved Figure 14 to include dataset labels.
1. Clarification on how the hyperparameters are chosen
   - Answer  provided in response to reviewer `pvM3` Q2, with extensive ablations conducted in Section 6.3.
   - Clarified that ThinKV does not require per-model or per-dataset tuning.
1. Clarification on SoTA methods
   - Answer provided in response to reviewer `pvM3` Q3.
1. Pareto-front of accuracy vs KV Cache size
   - Added new Figure 17 in Appendix E.11, demonstrating that ThinKV sets a new Pareto-frontier.

###  Reviewer `86j3`
1. ThinKV experiments on Qwen3
   - Added new results in Table 13 in Appendix E.13.
   - ThinKV in thinking mode incurs ≤2.2\% accuracy drop across different eviction budgets while using <6.87\% of the FullKV memory, compared to 33% accuracy drop of non-thinking mode.
1. QAQ-like quantization-sensitivity analysis
   - Added new Figure 16 and Appendix section E.9.
1. Fine-grained latency breakdown
   - Table 5 provides latency breakdown and presents average number of calls for each operator across decode steps.
   - Added new Figure 18 and Appendix E.14, showing latency breakdown analysis across batch sizes.
   - We additionally provide extensive latency e2e analysis in Figure 9, Appendix E.15 and Table 14.
1. Extension to long input-context workloads
   - Clarified that we treat prefill tokens as R-type thoughts.
   - Demonstrated that ThinKV is orthogonal to other prefill techniques by integrating with SnapKV.
   - Conducted experiments on LongBench v2 and added Appendix E.16 and Table 15.

We express our sincere gratitude to the reviewers for their careful assessments and constructive feedback, which greatly enhanced the clarity and rigor of this work.

Best regards,
The Authors

---

### Author Response · Authors · 2025-11-20
**General Response by Authors (Part 1 of 2)**

Dear Reviewers, ACs, SACs and PCs,

We sincerely appreciate your time and effort in overseeing the review process, and we value the thoughtful and constructive feedback provided by all reviewers, as well as the recognition of the novelty and contributions of our work. In the sections below, we summarize the strengths identified by each reviewer, summarize our responses to the concerns raised—each of which helped further strengthen our contributions—and outline the revisions incorporated into the manuscript.

## Strengths Recognized by Reviewers

### **Novelty of Contributions and Motivating Analyses**

- **Reviewer `P5bA`** — Emphasized the depth of the paper’s analysis and the novelty of deriving thought-adaptive KV compression from attention sparsity, noting the strong methodological insight behind ThinKV’s design.
- **Reviewer `ESb6`** — Pointed out that the work demonstrates why different tokens across reasoning trajectories benefit from different compression levels, and acknowledged the value of combining quantization with eviction.
- **Reviewer `86j3`** — Highlighted the innovation in identifying thought type from attention sparsity and using it to guide both quantization and eviction within a unified compression framework.
-  **Reviewer `pvM3`** — Underscored the relevance of the problem being addressed and noted that exercising ThinKV ultimately translates into smarter models.

### **Acknowledgment of Comprehensive Design**

- **Reviewer `P5bA`** — Praised the operator-level implementation.
- **Reviewer `86j3`** — Appreciated the thorough system breakdown, kernel-level analysis, and the extension of PagedAttention to support efficient reuse of evicted slots.

### **Validation of Empirical Performance**

- **Reviewer `P5bA`** — Highlighted that ThinKV achieves >95% accuracy even with 20–40× compression and delivered substantial throughput gains (up to 5.8×) and larger batch sizes compared to R-KV and H2O.
- **Reviewer `ESb6`** — Noted that the experiments are conducted on state-of-the-art reasoning models across challenging math and coding tasks.
- **Reviewer `pvM3`** — Observed that ThinKV offers better accuracy–efficiency tradeoffs than SoTA KV compression approaches.
- **Reviewer `86j3`** — Acknowledged that ThinKV consistently outperforms competing methods across memory budgets and provides both accuracy and throughput benefits.

---

### Author Response · Authors · 2025-11-27
**Gentle Reminder of Discussion Phase Nearing Completion**

Dear Reviewers,

As we approach the conclusion of the discussion phase, we would like to express our sincere gratitude to all reviewers for your time, thoughtful feedback, and insightful questions. We greatly appreciate your engagement in the discussion, and for indicating that our clarifications have addressed your concerns. The positive updates to your scores and evaluations are truly appreciated.

If any further questions arise or if additional clarifications would be helpful, we are fully available and happy to respond promptly.

For reviewers who have not yet had the chance to participate in the discussion, we would greatly appreciate any feedback you may be able to provide. Your perspective is important for a comprehensive assessment of the work and would meaningfully contribute to strengthening the paper.

To briefly reiterate, our submission **ThinKV**:

1. **Identifies, for the first time, that degrees of attention sparsity naturally enable dynamic thought decomposition** in reasoning models.
2. **Introduces a hybrid, thought-adaptive quantization–eviction framework** that assigns precision and performs eviction based on thought importance and the progression of the chain of thought.
3. **Proposes Continuous Thinking (integrated in vLLM)**, enabling efficient in-place reuse of KV memory without costly gather-based compaction—eliminating a major system bottleneck in prior approaches.

We believe these contributions will help guide future research in LRMs and beyond, while enabling practical performance improvements in real-world frameworks.

Thank you again for your time and effort in reviewing our paper.

Best regards,
Authors of` ''ThinKV''

---

### Meta-Review · Area_Chair_t5b7 · 2026-01-11

**Summary:**

This paper introduces ThinKV, a thought-adaptive KV cache compression framework that combines quantization and eviction tailored to different reasoning stages (reasoning, execution, transition) identified via attention sparsity patterns. The key insight that different thought types warrant different compression levels is well motivated and novel. Strengths include the unified hybrid quantization-eviction design, strong empirical results (near-lossless accuracy with <5% KV cache, up to 5.8x throughput), comprehensive ablations, and solid system-level implementation extending PagedAttention. Initial weaknesses included concerns about whether the approach was sufficiently principled versus an engineering recipe, limited end-to-end latency evaluation in serving frameworks, and lack of evaluation on long-input workloads. The rebuttal comprehensively addressed these concerns with vLLM integration, additional experiments, and clarifications on the principled sparsity-driven formulation.

**Reviewer Concerns:**

The authors addressed all major concerns satisfactorily. They integrated ThinKV into vLLM and provided comprehensive end-to-end latency evaluations showing 33% lower latency and 38% higher throughput under dynamic serving. They added detailed operator ablations (Table 4) isolating TBQ and TBE contributions. They clarified that the method is grounded in principled sparsity-driven formulation with extensive ablations justifying design choices. They added experiments on Qwen3, long-input workloads (LongBench v2 with SnapKV integration), QAQ-style quantization sensitivity analysis, and Pareto-front plots. All four reviewers explicitly acknowledged their concerns were addressed in the discussion.

**Reviewer Scores:**

P5bA (8): Maintained score, explicitly stated rebuttal addressed all questions. 86j3 (6): Acknowledged concerns were well addressed; score likely unchanged or slightly higher. ESb6 (4→6): Raised score from 4 to 6 after rebuttal, stating all concerns resolved. pvM3 (6): Did not yet respond post-rebuttal but concerns were addressed with updated figures, ablations, and Pareto analysis; score likely unchanged.

---

### Decision · Program_Chairs · 2026-01-26

Accept (Oral)